environmental science/applied mathematics/ statistical physics

accessibility, science of city, isochrone, public transport, urban planning

**Author for correspondence:**
Indaco Biazzo
e-mail: indaco.biazzo@polito.it

# General scores for accessibility and inequality measures in urban areas

Indaco Biazzo[1], Bernardo Monechi[2] and Vittorio Loreto[2,3,4]

[1]Politecnico di Torino, Corso Duca degli Abruzzi 24, Torino, Italy
[2]SONY Computer Science Laboratories, Paris, 6, rue Amyot, 75005 Paris, France
[3]Complexity Science Hub, Josefstädter Strasse 39, 1080 Vienna, Austria
[4]Physics Department, Sapienza University of Rome, Piazzale Aldo Moro 2, 00185 Rome, Italy

 IB, 0000-0002-9897-7543

In the last decades, the acceleration of urban growth has led to an unprecedented level of urban interactions and interdependence. This situation calls for a significant effort among the scientific community to come up with engaging and meaningful visualizations and accessible scenario simulation engines. The present paper gives a contribution in this direction by providing general methods to evaluate accessibility in cities based on public transportation data. Through the notion of isochrones, the accessibility quantities proposed measure the performance of transport systems at connecting places and people in urban systems. Then we introduce scores ranking cities according to their overall accessibility. We highlight significant inequalities in the distribution of these measures across the population, which are found to be strikingly similar across various urban environments. Our results are released through the interactive platform: www.citychrone.org, aimed at providing the community at large with a useful tool for awareness and decision-making.

## 1. Introduction

The inherent complexity of the emerging challenges human beings collectively face requires a deep comprehension of the underlying phenomena in order to plan effective strategies and sustainable solutions. Cities stand as a paramount example of how a complex interplay of infrastructures, technologies and human behaviours may lead to outcomes and patterns very far from the usual cause–effect scheme [1]. The science of cities is a research area that greatly benefited from the digital revolution in the last decades [2]. Nowadays, the deployment of information and communication technologies [3] and the consequent availability of an unprecedented wealth of data is opening new opportunities for a

scientific investigation into the complexity of urban environments. This availability of data fostered studies aimed at identifying the patterns of coevolution of human and social behaviours [4–7] as well as innovation at the level of infrastructures and services [8–12]. This paper aims at contributing to the ongoing debate about the future of our cities and the way to combine growth [13] with efficiency and inclusiveness. To this end, we focus on a specific aspect of cities, namely the topic of accessibility. Accessibility can be described as the capacity of cities to allow people to move efficiently by guaranteeing equity and equal access to personal and professional opportunities. From this perspective, accessibility does not mean only the overall capacity of urban transit: it also needs to be inflected as the accessibility of specific areas, for particular people with specific purposes. It is not rare that public transport projects spearheaded by governments benefit only a tiny fraction of the population, and while the average travelling conditions remain poor for the majority of the population. It is thus important to be able to quantify accessibility in a way that closely represents the experience of citizens. Following a common approach in accessibility studies [14], we focus on travelling times between geographical areas which better represents the mindset citizens adopt in planning their mobility. The key mathematical notion used to quantify travelling times will be that of *isochronic maps*, i.e. maps showing areas related to isochrones between different points. Considering a geographical point, its isochronic map will be composed by isochronic contours marking regions reachable in a given time-span, using different transportation systems. Isochronic maps have existed since 1881, when Sir Francis Galton published the first isochronic map in the *Proceedings of the Royal Geographical Society* [15], showing travel times in days from London to different parts of the world. Nowadays, the availability of data related to mobility allows for the compilation of very accurate isochronic maps for different locations, different geographical areas, different social communities and different transportation systems.

Though the notion of isochrone is well defined, its computation depends on the transportation system adopted. Here, we focus on public transportation, and we compute travelling times and isochrones using a routing approach that exploits multi-modality. This implies that the best route between two points A and B in the city can be realized through a combination of several transportation means (walking, buses, metro lines, trains). For the sake of simplicity and without loss of generality, here we only consider the official public transport schedules for many cities in North America, Europe and Australia. Following a recent interesting trend in scientific research [16], we developed visualizations on maps of this body of information, as well as several metrics for accessibility, through the open CityChrone platform (www.citychrone.org). Data about real-time passages of public transport journeys or other public or private means of transport can be easily integrated into the platform as well.

Usually, studies about public transport analyse the networks of transport as static graphs, where the nodes represent stops and the edges represent the routes connecting them [17–21]. Very few studies have instead incorporated in a systematic way the 'temporal' features of these systems [5,9,22], i.e. how users navigate through urban networks to reach their destinations. Here, we focus specifically on the dynamical aspects of mobility and we introduce two general metrics for accessibility of cities: a *velocity score*, quantifying the overall velocity of access to a specific area of the city, and a *sociality score* that quantifies how many people one can meet from a specific area. Finally, the dependence of the sociality score on the total population of a city can be reduced by scaling it with this quantity. In this way, we define a third accessibility metrics called *cohesion score* that quantifies the fraction of the total population that can be met with a typical trip starting from the considered location. The metrics adopted are defined 'general' in the sense that they can be applied in every city and different context allowing comparison between different areas and means of transportation. The proposed metrics allow for an extensive study of the level of accessibility of urban areas, a concept formulated several decades ago and used in different contexts in the literature [22–26] to quantify the performance of transportation systems in relation to various aspects of individuals' lives.

There is not just a single definition of accessibility. Depending on the context, the term accessibility could refer to the availability of services for disabled or disadvantaged people [27], the capability of reaching workplaces for ordinary citizens [24], and the possibility of attending certain activities at given times during the day [28]. Similarly, accessibility can be focused on travelling times using all or several modes of public or private transport or can rely on the spatial distribution of commodities and venues [14]. This proliferation of definitions can make it difficult to reach a unifying view about cities and their dynamical aspects, contributing instead to a dispersion of scientific efforts in diverging directions. The lack of a comprehensive and easy-to-understand definition of accessibility could prevent policymakers from using it in an operational way and scholars from comparing different approaches and methodologies [14]. Our aim here is to contribute towards a unified and reproducible point of view. Thanks to state-of-the-art routing algorithms, our metrics are designed to be efficiently

computed in relatively short times (less than 1 min for medium-sized cities). This opens the possibility to explore different scenarios close to real time. Also, our metrics are well suited for being shared and easily visualized on maps, making them easy to be applied by other researchers to reproduce and extend our results.

The quantification of inequality in accessibility has been proven to be an important tool to assess economic and social inequalities at an extra-urban scale [29]. It is worth mentioning that the local nature of our metrics allows us to evaluate and visualize the geographical fluctuations of the velocity and sociality scores, and thus to quantify the inequalities distribution of these measures among areas and population within each city. In particular, we show that while the distributions of the accessibility metrics seem to have higher values for high-density areas, only a small fraction of the population lives in areas with accessibility scores much larger than the rest of the city. Moreover, the performances of public transport systems decrease in an exponential-like way for all the observed cities with the temporal distance from the city centre. These results exhibit strongly similar patterns among all the observed cities, suggesting the existence of similar causes behind the emergence of this phenomenon, that could range from morphological to socio-economic ones.

Despite the local character of the proposed metrics, their aggregation at an urban scale allows for a quantification of the global level of the performances of public services of a city. In this way, the aggregated velocity score, the '*city velocity*', represents the overall velocity allowed by the public transportation services. On the other hand, the aggregated sociality score, the '*city sociality*', quantifies the number of people possibly met in a standard trip in a given city. The aggregated cohesion score, the '*city cohesion*', roughly indicates how well connected a random pair of individuals are in a given city. We adopt these aggregations to rank cities according to public transport performance. We find that, while in general there are correlations between the positions of a city in the different rankings, there are also interesting fluctuations due to the complex interplay between public transport and the population density.

The outline of the paper is as follows. In the Methods section, we illustrate the main tools we adopt throughout the paper, specifically the notion of isochronic map. We review its definition, and we describe how it is adopted in this paper, including the data and the algorithms to compute it. Based on the computation of these maps, we introduce several accessibility metrics to quantify the efficiency of the public transportation systems and the opportunities provided to the citizens in terms of mobility. The Results section describes several synthetic scores to allow a ranking of cities according to their accessibility patterns. Besides an overall evaluation, we focus in particular on the inequalities of accessibility in cities with respect to their space–time distribution. Finally, we draw some conclusions and highlight interesting future directions.

# 2. Methods

## 2.1. Isochronic maps

The accessibility metrics proposed in this paper rely on the notion of *isochronic maps*. An isochronic map is composed by a set of isochrones centred in a given location $\lambda$. The isochrone $I(\tau, \lambda)$ is the contour of the area reachable from $\lambda$ in at most a time $\tau$ and the ensemble of the isochrones obtained for different values of $\tau$ compose the isochronic map of the location $\lambda$. A more complete definition includes not only the travel time $\tau$ but also the absolute starting time of the trip. In this way, one has $I(\tau, (\lambda, t_0))$ as the contour of the area reachable from $\lambda$ in at most a time $\tau$ starting at time $t_0$. Though the notion of isochrones has been explored at a quantitative level for a long time [30], it is possible nowadays to compute them massively and very efficiently, opening the possibility for insightful study. The computation of isochrones is based on the computation of the travelling times between any pair of locations in a city using a multi-modal approach that integrates the adoption of all the available public transportation means alternated with walking paths. In order to keep the computational times low, we adopted a hexagonal tessellation of the city area which still allows for an exhaustive representation of the public transportation services. We constructed a hexagonal grid with a side of hexagons of 0.2 km. It is worth noticing that not the whole area of a city is covered by hexagons.[1] We cover with hexagons all locations of a city containing at least a stop of the public service and all areas reachable from any stop of the public service with walking paths not longer than 15 min. In order to compute the walking paths between stops of the public service and the hexagonal grid, we use the back-end version of the

---

[1] We remark that a satisfactory definition of city and its extension is still lacking [2].

open-source routing machine (OSRM) [31]. The OSRM allows for the computation of shortest walking paths on the urban networks of each city, using the corresponding OpenStreetMap [32] network. As for the schedules of public transit, we relied on data released by public transport companies. Google adopted the GTFS standard file (https://developers.google.com/transit/gtfs/) to encourage public transport companies to release their data in a uniform way in order to be included in its map platform. It is nowadays possible to find hundreds of companies having released their data, and there are portals where this data is collected and exposed [33]. The databases of public transportation systems are strongly heterogeneous across cities. In some cases, some transportation means could be missing while other extra-urban ones could be included. For instance, for Berlin and London, the GTFS (general transit format system) data include all regional trains [33]. To use a unique and general criterion about the inclusion of areas and transportation means, we adopted the OECD/EU definition of urban areas as *functional economic units* [34]. The OECD/EU definition exploits the population density to identify an urban core (city core) and travel-to-work flows to identify the hinterland whose labour market is highly integrated with the core (commuting zones). With this definition in mind, we filtered out all the services lying outside both the cores and the hinterland regions from our tessellations. In addition to the database of public transportation systems, we used the population density data on coarse-grained to squares with a surface of $1\,km^2$. In order to match the smaller size of the hexagons (approx. $0.1\,km^2$) with the size of the square for the population density, we divided the population of each square among the overlapping hexagons proportionally to the fraction of overlapping surface. Data about population densities in urban areas have been gathered through the Eurostat Population Grid [35] for the European cities [36] and the Gridded Population of the world made by the Center for International Earth Science Information Network [37].

The final step to compute the isochronic maps is to put together the coarse-grained representation of a city with the schedule of its public transportation system and to compute travelling times between any pair of hexagons of the tessellation at different times of the day and/or different days of the week. The need for fast commercial transit services has fostered the development of many routing algorithms, capable of computing the optimal routes in urban environments and integrating many different transportation means. Many of these algorithms can perform 'multi-criteria' optimization, i.e. they can compute the optimal routes minimizing travelling times but also the number of vehicle changes or putting constraints on the arrival times [38,39]. For our purposes, we adopted a modified version of the *Connection Scan Algorithm* (CSA) [40], that we call the Intransitive Connection Scan Algorithm (ICSA). The exact formulation of the algorithm is described in electronic supplementary material, S1. At a basic level (i.e. not considering walking paths), the CSA features a computation time that scales linearly with the number of connections, i.e. displacements between any two stops of the scheduled public service. The CSA algorithm imposes substantial limitations on the walking path to move from one stop to another. These limitations do not allow its use in a real scenario in urban contexts. Our generalization of the CSA algorithm overcomes these limitations and considers walking paths of less than 15 min when moving from one stop to another of the public service. Thanks to the ICSA algorithm, it is possible to compute all the shortest-time-paths connecting the centres of any pair of hexagons in the tessellation at several starting times for a typical day of the week. For a typical city with $\approx 10^4$ hexagons, one needs to compute $\approx 10^8$ shortest paths. Each one of these shortest-time-paths will consider all the possible means of transport between two hexagons, including the possibility to move on foot to nearby hexagons to access the public transport service places within a given area. The corresponding computational times range between less than 2 min for a medium-size city (like Rome) and about 30 min for a big city (for instance, New York) on a single CPU of a standard personal computer. The algorithm is easily parallelizable, and it allows to use of the framework described here to implement planning tools where accessibility metrics can be computed in nearly real time (less than 1 min of computation). A Python implementation of the computation framework used is released open-source on Github https://github.com/CityChrone/public-transport-analysis.

## 2.2. Accessibility metrics

In this section, we introduce two universal scores of accessibility that allow for an easy comparison of different areas of the same city and different cities considered as wholes. Interactive representations of those metrics for a large number of cities are available at www.citychrone.org. The accessibility quantities proposed aim to measure the performance of public transport at connecting places (velocity score) and people (sociality score). Roughly speaking, the velocity score measures how fast it is possible to reach any point from any other point in the city. The sociality score measures the amount

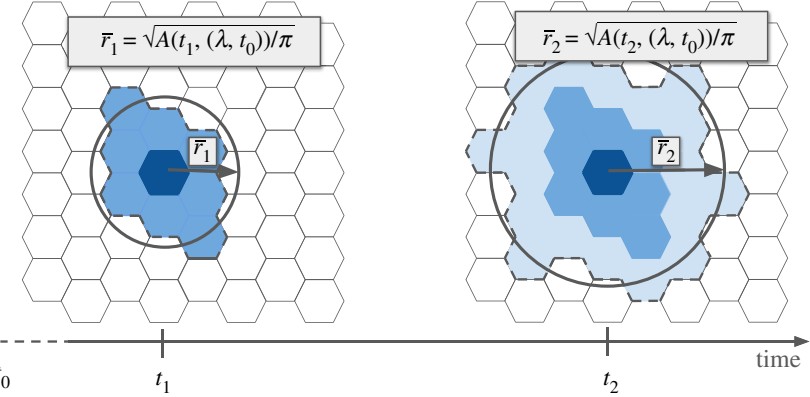

**Figure 1.** Isochrone area. Isochrones with hexagonal tessellation at different times. The circles in the figure have the same area as the area contained by the isochrones.

of population that it is possible to reach from any point in the city. Usually, the flow of people in urban systems is described by an origin-destination matrix (ODM). The velocity score can be thought of as an accessibility measure that assumes a uniform ODM. Conversely, in the case of the sociality score, we assume an ODM proportional to the population.

### 2.2.1. Velocity score

The *velocity score* aims at giving a synthetic representation of the information encoded in all the isochronic maps computed from all the points of a city. To this end, we imagine the isochronic map as a spreading process from a starting point, and we are interested in the average speed of expansion of the front of the isochrone as a function of time. More precisely, let us consider the isochrone centered at the hexagon $\lambda$ at time $t_0$ corresponding to a travel time $\tau$, $I(\tau, (\lambda, t_0))$. The *covered area* $A(\tau, (\lambda, t_0))$ of the isochrone at time $\tau$ will thus be the area contained within $I(\tau, (\lambda, t_0))$. By approximating the perimeter of the isochrone with a circle, the average travelled distance $\bar{r}$ taking a random direction from the starting point $p_0$ is given by

$$\bar{r}(\tau, (\lambda, t_0)) = \sqrt{\frac{A(\tau, (\lambda, t_0))}{\pi}}, \tag{2.1}$$

and dividing by the time $\tau$ we obtain a quantity that has the dimension of a speed

$$\bar{v}(\tau, (\lambda, t_0)) = \frac{\bar{r}(\tau, (\lambda, t_0))}{\tau}. \tag{2.2}$$

The interpretation of $\bar{v}(\tau, (\lambda, t_0))$ is the average speed of expansion, at time $\tau$, of a circular isochrone with the same area as the real one (figure 1).

This quantity can be considered approximately as the average velocity of a journey of duration $\tau$ choosing a random direction from the starting point. On the other hand, this quantity is proportional to the square root of the amount of area it is possible to explore from the hexagon $\lambda$ given a time interval of $\tau$. We chose to consider the square root of the area instead of the area itself to have a more direct interpretation of it in terms of transportation velocity, because it is easier to communicate and to understand for a general audience. This quantity is defined for every hexagon $\lambda$ and any starting time $t_0$ and travel time $\tau$. The velocity score is obtained by averaging over both the starting time $t_0$ and the travel time $\tau$, as

$$v(\lambda) = \frac{\sum_{t_0=06.00}^{22.00} \int_0^\infty v(\tau, (\lambda, t_0)) f(\tau) \, d\tau}{\sum_{t_0=06.00}^{22.00} \int_0^\infty f(\tau) \, d\tau}, \tag{2.3}$$

where several starting times have been considered, from 06.00 to 22.00 with a step of 2 h. In equation (2.3), the average over $\tau$ is performed by weighting with a travel-time distribution $f(\tau)$. The travel-time distribution represents the probability for an individual or a group of individuals to perform a journey of duration of $\tau$. The travel-time distribution could vary between the considered cities, time frames [14], and also between areas and groups of individuals of the same city [41]. In electronic supplementary material, S1, we show how the velocity score (and the other accessibility metrics defined in the following) computed with different choices for $f(\tau)$ are highly correlated with one

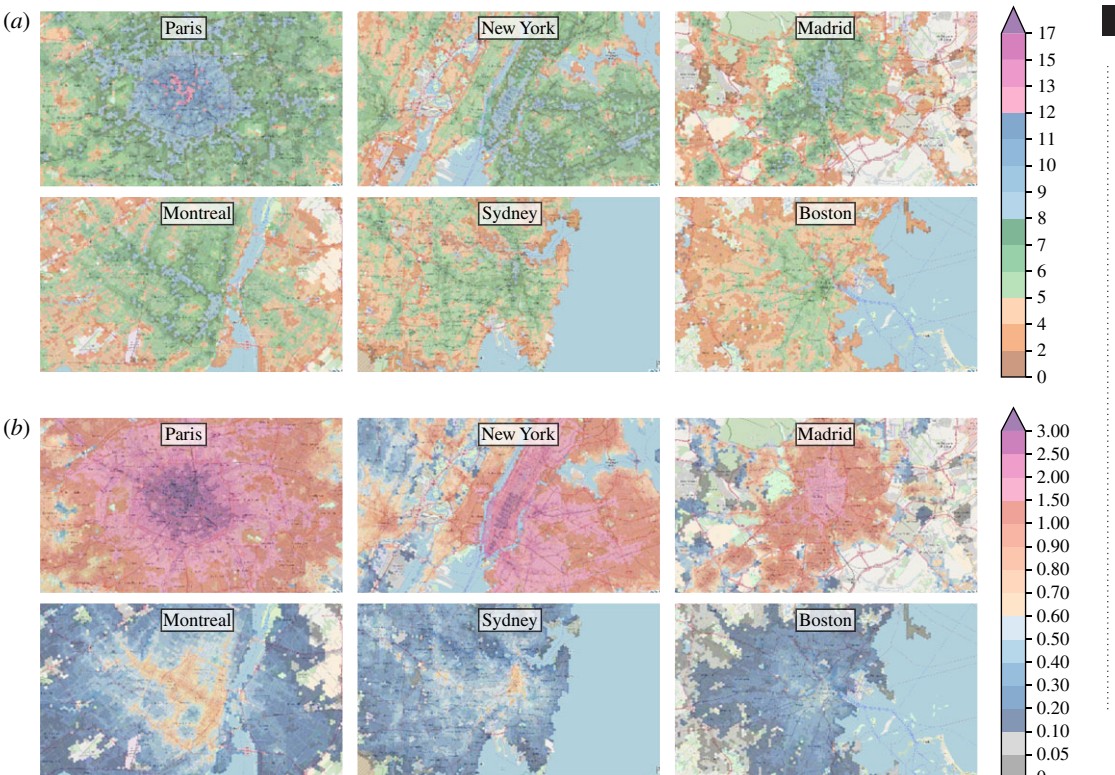

**Figure 2.** Maps of the velocity score and the sociality score. In the six maps of panel (*a*) we report the velocity score in km h$^{-1}$ and in (*b*) the sociality score in millions of inhabitants for six different cities: Paris, New York, Madrid, Montreal, Sidney and Boston. The values of the velocity score range from less than 2 km h$^{-1}$ (brown) of velocity score up to more than 17 km h$^{-1}$ (purple), whereas the *sociality score* ranges from less 0.05 millions of inhabitants reachable up to more than 3 millions of individuals. The great variability of the colours reveals a strong dissimilarity of performances of the public transport across cities.

another. Thus, the choice of $f(\tau)$ does not alter qualitatively the results obtained. On the other hand, using the same $f(\tau)$ for each city is equivalent to focusing on the perspective of a single individual, or a cohesive group of individuals, who would compare different cities and different transportation systems from their perspective. For all these reasons, we focused on one specific travel-time distribution, namely that obtained from fits of surveys of the daily budget times spent on a bus by UK citizens [6]. We remark that, though out of the scope of the present paper, the investigation of the impact of different city-specific travel-time distributions deserves further investigation.

Figure 2*a* shows the velocity scores of six different cities. For interactive explorations of the maps and other cities, we refer the reader to the platform www.citychrone.org.

## 2.2.2. Sociality score

The velocity score introduced above represents an indicator of how good the public service is at allowing a fast exploration of the urban space. At this stage, this score does not take into account the population density distribution. We know instead that there is a strong interplay and feedback loop between the efficiency of the public service and the population density. While it is normal to strengthen the service in highly populated areas, regions with a low population density risk being poorly served by public transportation. In order to quantify this interplay, we introduce a second metrics that quantifies the performance of public transit in connecting people. Let us now define $P(\tau, (\lambda, t_0))$ as the amount of population living within the isochrone $I(\tau, (\lambda, t_0))$. Similarly to what we did for the velocity score, we can average $P(\tau, (\lambda, t_0))$ over the travel time $\tau$ (with the same distribution of daily budget times $f(\tau)$) and over different starting times $t_0$, obtaining the sociality score as

$$s(\lambda) = \frac{\sum_{t_0=06.00}^{22.00} \int_0^\infty P(\tau, (\lambda, t_0)) f(\tau)\, d\tau}{\sum_{t_0=06.00}^{22.00} \int_0^\infty f(\tau)\, d\tau}, \tag{2.4}$$

Considering a typical working day, the velocity score provides an approximate measure of the average speed

at which an individual can move away from a hexagon $\lambda$, in a randomly chosen direction. Instead, the sociality score provides a measure of the number of people it is possible to reach within the same trip. The sociality score can also be interpreted as a measure of the amount of the population that can easily reach the point considered, assuming that, on average, the travel time of trips in cities is similar reversing origin and destination. In order to validate this assumption, we compute the sociality score with travel time of the incoming trip and outgoing trips for each point in Rome. In electronic supplementary material, figure S5, there is the scatter plot of these two quantities showing the high correlation between these two measures. Then figure 2b shows the sociality score maps for the same cities considered for the velocity score.

# 3. Results

## 3.1. City rankings

The scores introduced above allow us to rank cities according to the overall performances of their public transport system. To this end, we introduce the *city velocity* indicator as the average velocity score, weighted over the population density. The second indicator, we introduce is the *city sociality*, defined as the average sociality score weighted over the population density. While the city velocity is a measure of the how fast a typical inhabitant can visit the city on a typical trip, the city sociality is a measure of the how many distinct people it is possible to meet. Finally, we introduce the *city cohesion* indicator, which measures the easiness for two randomly picked individuals to meet within a city. The larger this indicator is, the more the city is cohesive and favours social interactions among its citizens. We note that the assumption of using the same travel-time distribution $f(\tau)$ for each city is quite strong since citizens of different cities might exhibit different travel habits. However, we are focusing on the perspective of a single individual, or cohesive group of individuals who would compare different city, as explained in the subsection Accessibility metrics. In electronic supplementary material, S1, we show how the rankings weakly depend from a reasonable choice of travel-time distributions.

### 3.1.1. City velocity

For each hexagon, $\lambda$, we have both the number of people living there, pop($\lambda$), as well as the average velocity of their trips with public transport starting from the considered hexagon, $v(\lambda)$ (equation (2.3)). In this way, we can compute the average velocity per person of the whole city, representing the average amount of different places a typical person living in the city can easily access with public transit. In particular, we define the city velocity as the average velocity per person

$$v_{\text{city}} = \frac{\sum_{\lambda \in \text{city}} v(\lambda) * \text{pop}(\lambda)}{\text{pop}(\text{city})}, \tag{3.1}$$

where pop($\lambda$) is the population in the hexagon $\lambda$. In equation (3.1), we sum over all the hexagons in the city weighted by the population living in that hexagon, and we divide by the total of the population of the city (living in the core and the commuting zones), pop(city). Note that we assign zero velocity to all the areas of the city not covered by hexagons, i.e. the areas more than 15 min away from any stop. Figure 3 reports the ranking of several cities according to their city velocity. The highest-ranked cities are Berlin and Paris, with values 20% higher than any other city. This means that typically a citizen of Berlin and Paris can explore the space around at least 20% faster than the others. Copenhagen, Helsinki, Athens, Prague, London and New York feature good performance. On the other side of the spectrum, Mexico City, San Diego and other US cities have a large fraction of the population with very low-velocity score.

### 3.1.2. City sociality

The city sociality is defined as

$$s_{\text{city}} = \frac{\sum_{\lambda \in \text{city}} s(\lambda) * \text{pop}(\lambda)}{\text{pop}(\text{city})}. \tag{3.2}$$

As for the city velocity, we average over the population distribution, and the areas of the city not served by public transport are considered to have zero sociality score. The city sociality is the typical number of

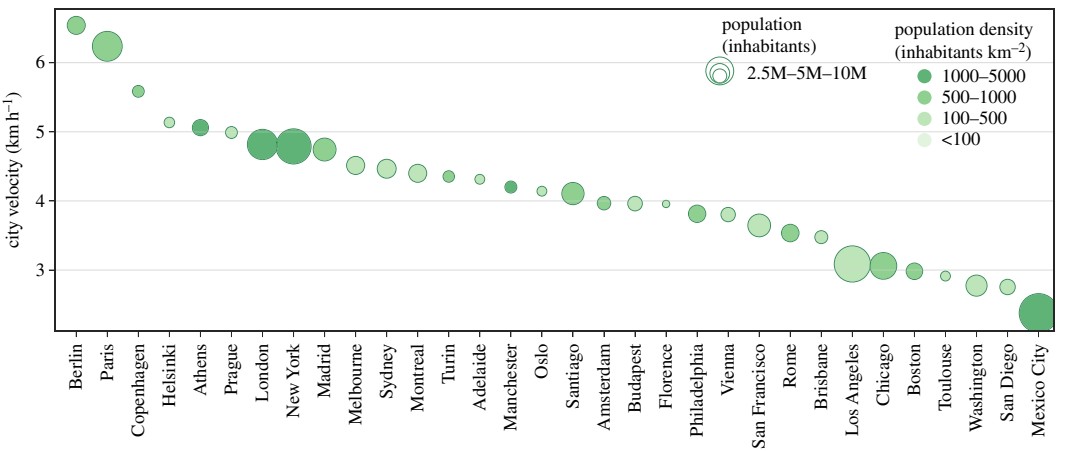

**Figure 3.** Ranking of cities according to the city velocity defined in equation (3.1). Cities are displayed with circles whose size is proportional to the total population and whose saturation of the filling colour is proportional to the overall population density.

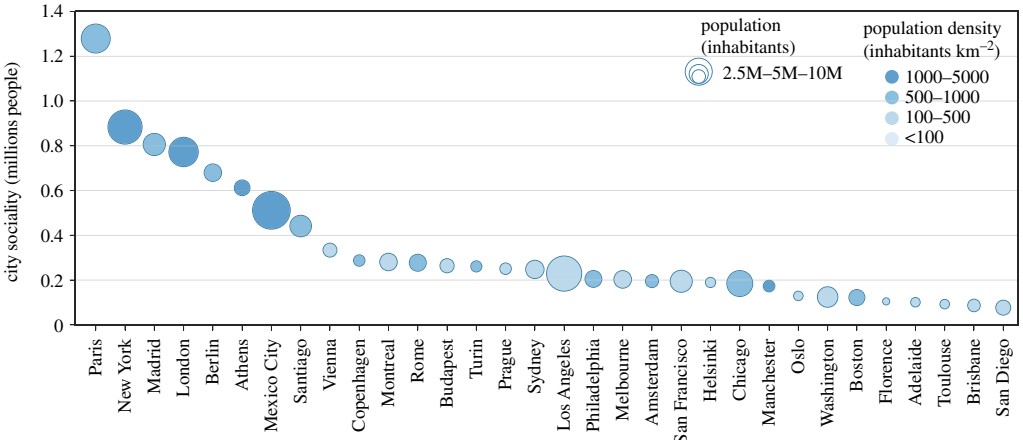

**Figure 4.** Ranking of cities according to the city sociality defined in equation (3.2). Cities are displayed with circles whose size is proportional to the total population and whose saturation of the filling colour is proportional to the overall population density.

people that a person living in the city can potentially meet within a typical daily trip. The ranking of cities, according to the city sociality, reported in figure 4, features some differences for the corresponding ranking obtained with the city velocity. In this case, Paris gains the first position thanks to its high population density in the city core and its efficient and capillary public transit system. Among the set of considered cities, Paris is the only one where on average a person can potentially meet over one million people in a typical daily trip. Scrolling the ranking, the city sociality decreases initially quickly, with the most populated cities in the first positions, then eventually decreases very slowly for smaller cities.

### 3.1.3. City cohesion

By re-scaling the city sociality with the total population of a city, we obtain the city cohesion

$$c_{\text{city}} = \frac{s_{\text{city}}}{\text{pop(city)}}. \tag{3.3}$$

The city cohesion gives an estimate of the fraction of the population that can be reached by a typical trip of an inhabitant of the city. Figure 5 shows the ranking of cities according to their city cohesion. The first city is Athens, thanks to a good public transportation system and a very high-density population concentrated in the core of the city. In second and third positions are Berlin and Copenhagen, which also feature very high velocity scores. Then we find Turin and Florence featuring a good balance between the population distribution and the efficiency of the public transportation system, despite

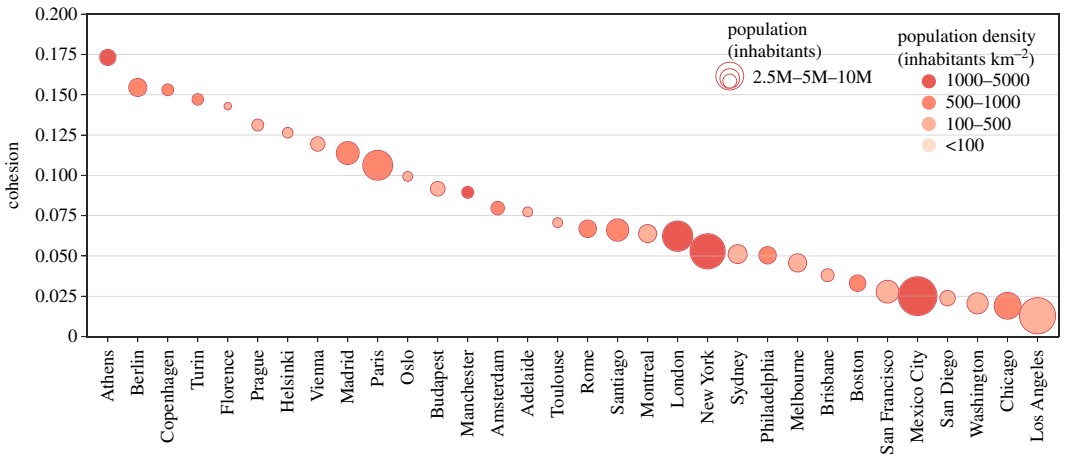

**Figure 5.** Ranking of cities according to the city cohesion defined in equation (3.3). Cities are displayed with circles whose size is proportional to the total population and whose saturation of the filling colour is proportional to the overall population density.

relatively low city velocity and city sociality. A large proportion of US cities have a low city cohesion score, resulting from the low population density in the city core, making those cities very dispersive.

## 3.2. Inequalities in urban accessibility patterns

In this section, we focus on a particular aspect of accessibility, the spatial–temporal distribution inside the city. A high position of a city in the overall ranking for any of the scores presented above does not imply *per se* that the same accessibility patterns are granted to all citizens. In order to investigate dis-homogeneities in the accessibility patterns, one needs to take a closer perspective and look at the accessibility metrics at a more fine-grained scale within cities. We focus for visualization clarity reasons on a subset of cities, namely the same cities we focused on in the section devoted to Accessibility metrics: Paris, New York, Madrid, Montreal, Sydney, Boston. In electronic supplementary material, S1, we show how the results presented are valid also for the other cities analysed.

An interesting way to represent the velocity and sociality score is through a violin plot, as reported in figure 6. Panels *a* and *b* refer to the distributions of the velocity and sociality scores, respectively. The way in which one reads these plots is the following. For each city, we plot the distribution of areas and population as a function of the velocity or sociality score. For instance, panel *a* refers to the velocity score. For each city, we plot in light green the normalized distribution of areas (hexagons) as a function of the velocity score, i.e. the fraction of hexagons featuring a specific value of the velocity score. A very efficient city has this distribution peaked around high values of the velocity score. From this perspective, New York appears to have the most balanced distribution of velocity scores across its whole area. On the other hand, represented in dark green is the distribution of the population density as a function of the velocity score, i.e. the fraction of the population associated with a specific value of the velocity score. A city with well-distributed public transport accessibility among the population has this distribution peaked around high values of the velocity score. From this perspective, Paris, New York and Madrid appear to have more equally distributed velocity scores than Montreal, Sydney and Boston. Panel *b* reports the same information as panel *a* (in light and dark blue) for the sociality score. The difference between Paris, New York and Madrid, on the one hand, and Montreal, Sydney and Boston, on the other, in terms of the range of sociality score both for areas and population is striking. Paris and New York appear to feature the broadest distribution of sociality scores across their citizens.

It is evident, both in panels *a* and in *b*, that (light green and blue areas) a large number of hexagons within the city borders display low values of the accessibility scores. However, when the population density is taken into account (dark green and blue areas), the peaks of the distribution shift towards high-populated areas. This result is somehow unsurprising, considering that the public transportation systems are mainly designed to serve the largest amount of citizen as possible as allowed by the limited financial resources. Figure 6 confirms this picture, where it is evident that there is a growing trend of the average values of the velocity scores at fixed population density with the population density for the six cities considered above. The trends reported above, correlating denser populated areas to higher (on average) accessibility scores, do not imply that the planning of public

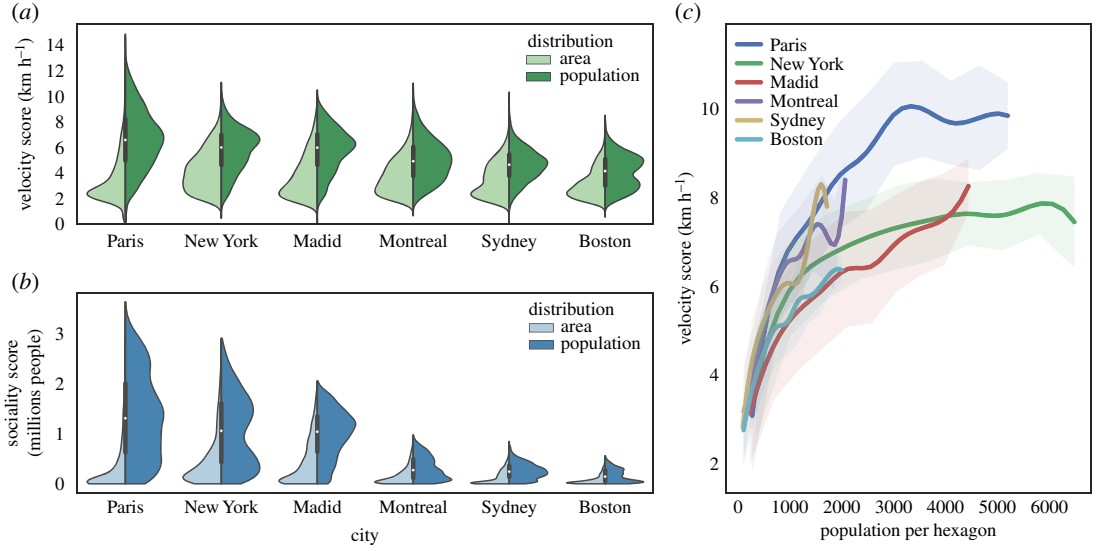

**Figure 6.** Distributions of the velocity (*a*) and the sociality (*b*) scores and velocity score versus the population density (*c*). (*a*) Distribution of the velocity score. The light green area represents the distribution of the areas featuring a given value of the velocity score. The dark green area represents the distribution of population with a given value of the velocity score. (*b*) Distribution of the sociality score. The light blue area represents the distribution of the areas featuring a given value of the sociality score. The dark blue area represents the distribution of population with a given value of the sociality score. The distribution for the population density is peaked towards higher values than that related to the area, signalling the fact that denser areas are associated, on average, to better public transit systems. We do not report here the results for the Cohesion metrics since it would give the same information of the sociality score. (*c*) The average value of the velocity score in hexagons with a given population for the six cities of Paris, New York, Madrid, Montreal, Sydney and Boston. In all cases, one observes an increasing trend. The shadows around the average values curve represent the standard deviation of the velocity score distribution for each corresponding value of the population

transportation systems succeeds in reducing inequalities in the accessibility patterns. The spread of the distributions is still very high and very few people (or areas) have access to high accessibility values compared to the rest of the populations (or areas). This is true for all the accessibility scores introduced above. In order to better quantify the large variability of urban accessibility patterns, we divide urban areas (hexagons) and population in two classes: hexagons and people featuring the top 1% of values of the Velocity and the sociality scores and the remaining 99%. For each of the two classes, we compute the average values of the velocity and the sociality scores and we compare them. The results are reported in figure 7: panels *a* and *b* for the velocity score and panels *c* and *d* for the sociality score, panels *a* and *c* for the distribution of hexagons and panels *b* and *d* for the population densities. The striking, though perhaps not surprising, result confirms the strong level of inequalities observed for all the cities considered. The ratio between the average values of the scores of the two classes is always larger than two. This implies that focusing for instance, on the velocity score, the top 1% of the hexagons (populations) features values of the velocity score that are double the remaining 99%. In other words, 1% of the city areas allow for daily trips at twice the speed of the rest of the city, and 1% of the population can move around at least twice as fast as the rest of the population. Similar considerations hold for the sociality scores, which implies that 1% of the population potentially has access to twice the number of people as the rest of the population. The ratio between the values of the top 1% compared to the remaining 99% is similar across all considered cities (see electronic supplementary material, S1, figures 5 and 6), as witnessed by the error bars reported in figure 7. It is also interesting to observe that almost all the ratios between values of the average scores computed for the two classes (1% and 99%) lie between 2 and 4, suggesting the existence of general patterns of organization across very different cities and urban environments.

### 3.2.1. Space–time distribution of inequality in accessibility patterns

The quantitative assessment of the strongly uneven distribution observed in the accessibility patterns reported above can be further clarified by looking at the spatial distribution of the accessibility metrics. In the maps shown in figure 2 (and at www.citychrone.org), we observe a central area with

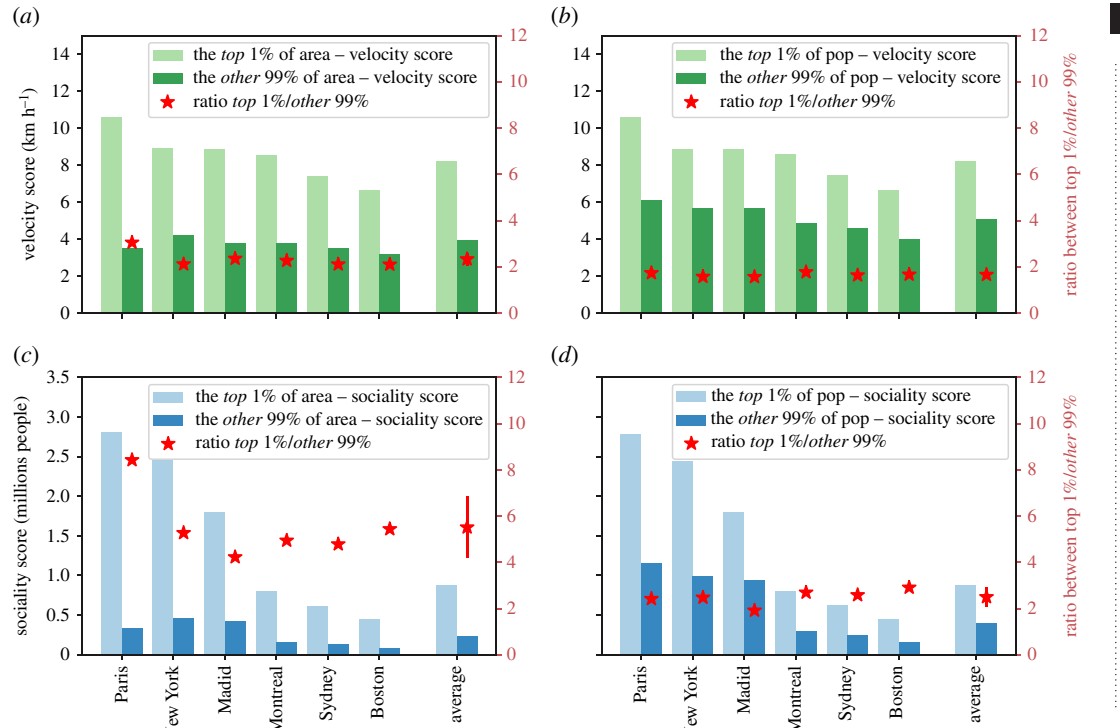

**Figure 7.** Inequalities in accessibility patterns. (*a*) Average values of the velocity scores among the hexagons featuring the top 1% (light green) of the values of the velocity score when compared with the remaining 99% (dark green) for the six selected cities. The last columns report the same values averaged over all the 32 analysed cities. Red star marks on the right *y*-axis are the ratios between the average values of top 1% and the other 99%. (*b*) Average values of the velocity scores among the population with the highest top 1% (light green) of the values of the velocity score when compared with the remaining 99% (dark green) for the six selected cities. The last columns report the same values averaged over all the 32 analysed cities. Red star marks on the right *y*-axis are the ratios between the average values of the top 1% and the other 99%. (*c*) Same as (*a*) for the sociality score. (*d*) Same as (*b*) for the sociality score.

the highest values of the accessibility observables and some 'islands' with high accessibility values connected to the central zone by some well-served directions, consistent with the idea of polycentric cities [42]. To better quantify this effect, we show the behaviour of the velocity and the sociality scores (figure 8) as a function of the travel time from the centre of each city. Here, the centre of a city is defined as the hexagon with the highest score (velocity and sociality, respectively). Both the velocity and the sociality scores decay fast as a function of the travel time from the city centre. The exponential function well describes this decay

$$f(t) = \sigma_0\, e^{-t/\tau} + \sigma_\infty, \tag{3.4}$$

where $\tau$ represents the typical decay time, $\sigma_\infty$ is the lower bound of the velocity score for each city, and $\sigma_0$ represents the average velocity score of areas (hexagons) nearby the best performing one. We performed the best fit by binning $\tau$ in order to remove biases coming from better sampled temporal distances. The value $\sigma_\infty$ represents the value of the score (either velocity or sociality) acquired in hexagons at the temporal edge of the city itself, i.e. for the farthest (in travel time) hexagons from the city centre. Hence, we estimated it as the average velocity score (sociality score) of the 5% least accessible hexagons from the city centre. The parameters $\tau$ and $\sigma_0$ are obtained through a linear regression of the quantity $\log (f(t) - \sigma_\infty)$, which depends linearly on $t$. The curve well fits the decay of the mean values of the velocity score, the average value among all 32 cities analysed of *R*-square is $R^2 = 0.92$ (see electronic supplementary material, S1, figures 7 and 8). The average value of the characteristic time $\tau$ is 0.86 h, ranging from 0.4 h for Santiago to the 1.6 h for Los Angeles. The dependence of the sociality score on the temporal distance from the best performing hexagon of the city is again well described by the equation (3.4), with a value of *R*-square $R^2 = 0.95$ higher with respect to those of the velocity score and an average value of $\tau = 0.55$ h. The smaller characteristic time is due to the convolution of the decay of the velocity score with the well-known decay of the population density from the city

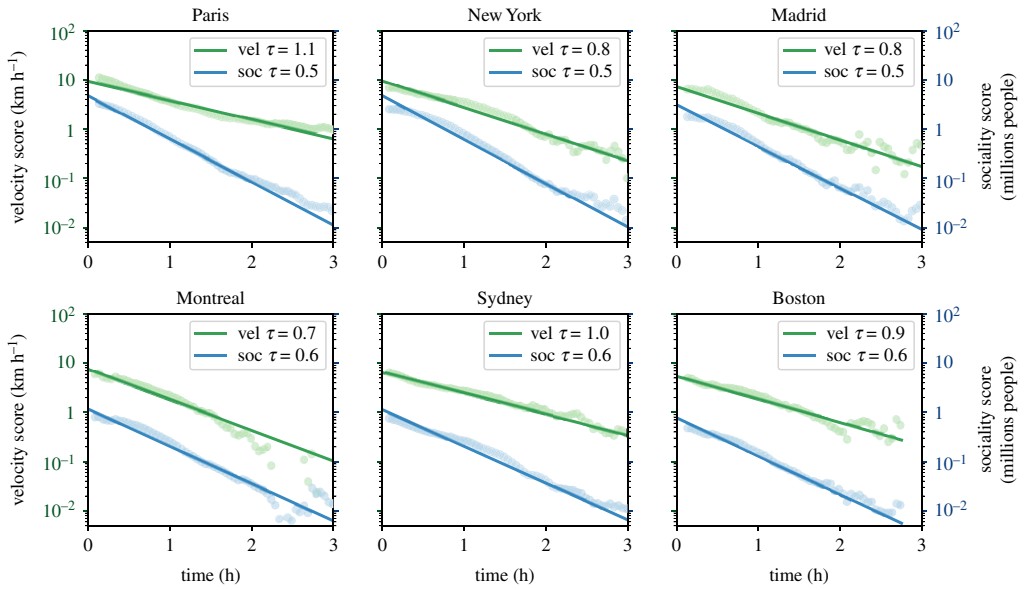

**Figure 8.** Exponential decay of the velocity score and the sociality score with travel times from the city centre. Average values of velocity scores (green points) and sociality scores (blue points) at a given travel-time distance from the hexagon with the highest score in each of the six selected cities. The lines are the best fit of the data with the function (3.4). In the legend for each figure, the parameters $\tau$ of equation (3.4) found for the decay of velocity score and the sociality score are reported.

centre, which is again exponential [43–45]. The ensemble of the results confirms that inequalities in the accessibility patterns allowed by public transport favour a small portion of the total city area and a small fraction of the population, typically clustered around certain areas. Moving in space and time away from these areas will lead to experiencing generally much less performing public transport services. This behaviour and the stability of inequality patterns of figure (7) strengthen the hypothesis of shared causes, independent from the particular location, behind the emergence of the decay of public transport performances that will deserve future investigations.

## 4. Discussion

The study of accessibility in urban contexts represents a multifaceted topic whose relevance transcends the mere problem of optimizing transportation systems, though this a very complex. It impacts the level of opportunities available in a city, the equal access to them and the level of inclusion of minorities. The interpretations that one can give to the notion of accessibility can take are numerous: from the planning of better and more efficient urban environments to the improvement of quality of life in rural areas, from the definition of real estate market prices to the definition of new business models of mobility and so on. The extreme generality of the term accessibility also depends on the specific aspects one could be interested in: the availability of jobs in a specific area, the quality of the schools in a neighbourhood, the possibility to take part in leisure activities depending on the time of the day. Despite a long tradition of scientific studies on these subjects, no consensus has yet emerged on how to quantify accessibility in a general way, i.e. through metrics applicable in very many situations and very different urban contexts. The main aim of this paper is to give a contribution towards a unifying, simple and general framework for accessibility studies. We proposed some general metrics that allow for a quantitative comparison of different cities and different areas of the same city. Despite the limitations of some of our assumptions, our framework and measures are easily reproducible and applicable to the study of accessibility via public transport in every urban environment in which transit feed open data is available. To this end, we took a specific angle by looking at the city and at the paths within it from the point of view of travelling times, which allows mapping the city in a way much closer to individuals' perception. The cornerstone of this approach is the computation of isochronic maps and, based on them, the introduction of several scores that take into account the performance of public transport to connect areas and people. The primary outcome is a set of scores that quantify how well a city is served by the public transit and how well a specific area of the city is connected to the rest of the city. We show how these scores allow comparing the performances of

public transport systems of different cities around the world, pointing out the differences in their ability to expand the range of opportunities and enhance social interactions. A very interesting opportunity opened by the new scores concerns the possibility to quantify the level of uneven distribution of these quantities within a city, i.e. the fluctuations of the accessibility scores among areas and population. We remark here that our first aim was to measure the performance of public transport to connect places and people. Despite that more realist origin destination matrix, for instance, considering the distribution of opportunities within the city, can be considered and easily integrated into our framework. But, up to now, there is still a lack of open datasets covering enough cities for this kind of analysis. Taking into account the aim of the accessibility measure proposed, our analyses reveal a general pattern observed in all the considered cities. Namely that 1% of the area of a city features accessibility scores with average values at least double those of the remaining 99% of areas. The same patterns are observed by looking at the number of people enjoying specific values of the accessibility scores: also in this case, the top 1% of the population can move at least twice as fast as the remaining 99% of the population. This very uneven distribution of performances of the public transport within an urban environment is explained in terms of the rapid decay of the accessibility scores as a function of the temporal distance from the city centre. The observed similarities of the mobility patterns across different cities suggest the existence of common causes, independently of the specific location. The observed inequality patterns are the results of the planning and organization of public transport systems. Considering our initial remarks, we can speculate that these patterns might be explained by the limited resource urban planners have to deal with when designing public services. In this sense, including important locations and fluxes might allow us to understand if these resources are efficiently allocated to satisfy the mobility needs of the citizens. The availability of general scores for accessibility and inequalities could be the first step towards a more systematic evaluation of the present situation in urban contexts and careful planning of future scenarios. The www.citychrone.org platform is a relevant example in that direction, already allowing for both the visualization of all the accessibility metrics introduced here and the conception of new scenarios for improved mobility and accessibility. As a final remark, the inclusion of other data sources, such as points of interest in the accessibility metrics (e.g. workplaces, shops, schools, etc.) or considering fluxes of people is quite straightforward in our framework and could lead to interesting results either in the global ranking of accessibility between cities and in the comparisons between city areas, by giving more importance to the purpose and popularity of certain trips.

Data accessibility. All the data used in this work can be freely accessed from public repositories [33–35,37]. Python source code used to compute the accessibility quantities and for the analysis performed are freely downloadable from the online *public-transport-analysis* GitHub repository (doi:10.5281/zenodo.1309835) [46]. The hexagons tessellation and the related accessibility quantities computed can be download from the online *openData* GitHub repository (doi:10.5281/zenodo.1309927) [47]. In the same repository, there is also a CSV file (agency.csv) with the list of public transport agencies used for each city.
Authors' contributions. I.B., V.L. and B.M. designed the research; I.B. performed the analysis; I.B., V.L. and B.M. analysed the results and wrote the paper.
Competing interests. The authors declare no competing interests.
Funding. The authors acknowledge support from the project KREYON, funded by the John Templeton Foundation under contract no. P51663. The authors acknowledge Sapienza University of Rome and ISI Foundation for the support in the management of the funding. This work has been partially supported by the SmartData@PoliTO center on Big Data and Data Science, and by Sony Computer Science Labs Paris.

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
