## [Reviewer comments · Royal Society Open Science]

Review History

RSOS-181315.R0 (Original submission)

Review form: Reviewer 1 (Liang Ma)

Is the manuscript scientifically sound in its present form?

No

Are the interpretations and conclusions justified by the results?

No

Is the language acceptable?

No

Is it clear how to access all supporting data?

Yes

Do you have any ethical concerns with this paper?

No

Have you any concerns about statistical analyses in this paper?

No

Recommendation?

Accept as is

Comments to the Author(s)

Overall, I think this is a very interesting paper. This paper makes a unique contribution in offering a new method in measuring temporal accessibility and its relevance with social equity. The platform created provides excellent examples of visualizations of temporal accessibility in several world cities and has potential implications for transport policies, particularly in transport equity. Other cities may easily create similar accessibility maps using the codes provided by the authors.

Review form: Reviewer 2 (Michiel Van Meeteren)

Is the manuscript scientifically sound in its present form?

Yes

Are the interpretations and conclusions justified by the results?

No

Is the language acceptable?

Yes

Is it clear how to access all supporting data?

Yes

Do you have any ethical concerns with this paper?

No

Have you any concerns about statistical analyses in this paper?

I do not feel qualified to assess the statistics

Recommendation?

Major revision is needed (please make suggestions in comments)

Comments to the Author(s)

One of the interesting things about the "open review" process at the Royal Society Open Science journal is the idea of an 'unmasked' peer review: where I can explicate my own academic personality as a background to my reviewer judgment. So to start with that, I am Dr. Michiel van Meeteren, lecturer in human geography at Loughborough University. The reason why I believe I was chosen as a reviewer to this paper is because I was a co-author on the following paper:

"Boussauw, K., Van Meeteren, M., Sansen, J., Meijers, E., Storme, T., Louw, E., Derudder, B. & Witlox, F. (2018). Planning for agglomeration economies in a polycentric region: Envisioning an efficient metropolitan core area in Flanders, *European Journal of Spatial Development*, 69.<http://doi.org/10.30689/EJSD2018:69.1650-9544>"

In this paper we combine accessibility analysis with isochrone maps for a practical application in public transport planning. In doing so we explicitly draw on the long established history of using these techniques in transport geography and related disciplines. I have to admit up front that the method proposed in this paper is more sophisticated as what we used, and I am actually quite enthusiastic about the indicators that this paper proposes. However, my disciplinary background is also a source of criticism in this review, in the sense that “the claims to newness” in this paper does disenfranchise the contributions from geography and geographers to the study of accessibility somewhat. Claims that this paper is “a first contribution in this direction” (abstract), that “studying cities in terms of travel time is a minoritarian view” (page 2, for geographers it is the default view). “very few people have studied the temporal features of these systems” (page 2). tie into this feeling. It might be new to complexity science, computer science or physics, but it is not new to geographers at all. This is something the authors acknowledge when they cite some of geography’s foundational works, references 21-29 contain some of the classic papers I would normally recommend. This resonates with a broader annoyance as data science, physics and complexity theorists have been ‘discovering the city’ as a topic of research. We as geographers welcome new perspectives and input to some of our established research topics, but we do like to have some acknowledgement of our historical contribution to a topic, a recurring complaint nicely elaborated in O’ Sullivan and Manson (2015).

The authors mention that “that the science of cities is a relatively new research area (P.2)”. I would make the case that it needs qualification. I’d say that the author of that claim, Michael Batty has been doing the science of cities for the whole of his long and distinguished career and that he himself was able to establish that career on a network of scholars in the 1960s (Barnes and Wilson 2014 provide historical perspective; and I did my little bit in Van Meeteren and Poorthuis 2018). In fact, accessibility analysis goes back to the collaboration of a geographer and a physicist on population potential in the 1950s (Stewart and Warntz, 1958). In other words, this reviewer would be very happy to see this contribution not as the “invention” of a debate/concept but as a “contribution/refinement” to a longstanding, but perhaps at certain times in history dormant research agenda. Most of this revision is just a question of language and due citation. Even nowadays there is quite a bustling community of geographers working on these topics, for instance just look at journals such as “Environment and planning B: urban analytics and city science” (edited by Mike Batty), The Journal of Transport Geography or Transportation Research. So although I agree with the authors that big data and massive computation offers great potential to revisit these topics (page 4), it is not that there is not a large community of scholars doing exactly that.

Which brings me to another comment: I don’t think this community of geographers has no standard measures of accessibility: we have (see Boussauw et al 2018 for references) and they are quite similar in type to what this paper proposes.

That said, I am very enthusiastic about the new measures “velocity score”, “sociality score” and “city sociality” and I like the crisp and unambiguous way in which the authors define it. Those measures will be a very welcome and useful tool for comparative research across cities both for academic and for practical applications. Geographers do not like describing indicators with adjectives like “universal” or “law” for all sorts of epistemological and historical reasons (Barnes 2013). Nevertheless I understand there is a problem of translation between disciplinary languages here.

To conclude there is one final weaker point in the paper I would like to point out. The section “Space-time distribution of inequality in accessibility patterns” (page 9-10) assumes a monocentric model of a city. This monocentric model has been increasingly found to be unrealistic of city structures regardless of where you are (Van Meeteren et al. 2016 is a good introduction to these

debates see also Clark and Kuijpers-Line 1994, Clark 2000). Of course there are car based and public transport based cities and North American cities tend to be the former for historical reasons. So the rankings provided do not surprise a geographer at all. They are exactly what one would expect based on geographical knowledge. However, if the paper is serious in its ambition to provide a cartographic tool that helps policy makers, it should take into account that cities have several employment centres and that the traditional 'urban core' has long since lost its meaning as the principal centre of employment public transport is directed toward. Most new jobs come in the city's edge (Phelps, 2017; Keil, 2017), therefore championing cities that have the best accessibility to the city center is not always the best policy advice, although advocating for higher densities that allow for public transportation surely is. On that note, there is a literature on 'threshold values' when public transport becomes viable. This is not so much a "general mechanism" (page 10) as the simple result of transport economics that require a minimal number of customers for an investment to be politically and economically sustainable. Again here we encounter some disciplinary loss in translation. For social scientists these "general mechanism" are the product of social and political decisions which makes us cautious in using terms like "general" and "universal" as society could easily change that reality if enough people want to.

Lastly, the 'most of the world now lives in cities' argument is now so much of a cliché (that hides a lot, see Brenner and Schmid 2014) that I would avoid using it.

I hope these points allow the author to improve on what is already a really interesting paper. I look forward to see it develop further.

Barnes, T. J. (2013). Big data, little history. *Dialogues in Human Geography*, 3(3), 297-302.

Barnes, T. J., & Wilson, M. W. (2014). Big Data, social physics, and spatial analysis: The early years. *Big Data & Society*, 1(1), 1-14.

Boussauw, K., Van Meeteren, M., Sansen, J., Meijers, E., Storme, T., Louw, E., Derudder, B. & Witlox, F. (2018). Planning for agglomeration economies in a polycentric region: Envisioning an efficient metropolitan core area in Flanders, *European Journal of Spatial Development*, 69. <http://doi.org/10.30689/EJSD2018:69.1650-9544>

Brenner, N., & Schmid, C. (2014). The "Urban Age" in Question. *International Journal of Urban and Regional Research*, 38(3), 731-755.

Clark, W. A. V., & Kuijpers-Linde, M. (1994). Commuting in restructuring urban regions. *Urban Studies*, 31(3), 465-483.

Clark, W. A. V. (2000). Monocentric to Policentric: New Urban Forms and Old Paradigms. In G. Bridge & S. Watson (Eds.), *A Companion to the City* (pp. 141-155). Malden/Oxford: Blackwell Publishing.

Keil, R. (2017). *Suburban planet*. John Wiley & Sons

O'Sullivan, D., & Manson, S. M. (2015). Do Physicists Have Geography Envy? And What Can Geographers Learn from It?. *Annals of the Association of American Geographers*, 105(4), 704-722.

Phelps, N. (2017). *Interplaces: An Economic Geography of the Inter-urban and International Economies*. Oxford University Press

Stewart, J. Q., & Warntz, W. (1958). Macrogeography and Social Science. *Geographical Review*, 48(2), 167.

Van Meeteren, M., Poorthuis, A., Derudder, B., & Witlox, F. (2016). Pacifying Babel's Tower: A scientometric analysis of polycentricity in urban research. *Urban Studies*, 53(6), 1278–1298. <http://doi.org/10.1177/0042098015573455>

Van Meeteren, M., & Poorthuis, A. (2018). Christaller and “big data”: recalibrating central place theory via the geoweb. *Urban Geography*, 39(1), 122–148.

Review form: Reviewer 3

Is the manuscript scientifically sound in its present form?

No

Are the interpretations and conclusions justified by the results?

Yes

Is the language acceptable?

Yes

Is it clear how to access all supporting data?

Yes

Do you have any ethical concerns with this paper?

No

Have you any concerns about statistical analyses in this paper?

No

Recommendation?

Major revision is needed (please make suggestions in comments)

Comments to the Author(s)

If my assumptions are correct, then the methodology is sound. However, the use of imprecise scientific terminology and the lack of rigour when explaining the mathematical equations often forces the reader to guess what the authors meant instead of what they wrote. This becomes particularly problematic when it results in masking some important limitations of the methodology. Although I believe that the following points should prove relatively easy to fix, I would strongly advise against publishing the manuscript until these are addressed, hence the major revision recommendation.

(1) L. 206: The speed of expansion of the isochrone is $dr/d\tau$, not r/τ . Please indicate the correct interpretation (l. 208-209) directly. In addition, the term “velocity” is generally used instead of “speed” when the direction is considered, while it has precisely been removed here.

(2) L. 209-210: v is certainly not proportional to the explorable area from the hexagon λ . If an isochrone moves at a constant speed of 1 m.s⁻¹ and is a perfect circle, then $v(1)=1$ and $A(1)=\pi$, while $v(2)=1$ and $A(2)=4\pi$, and more generally, $v(n)=1$ and $A(n)=n^2\pi$ for any n . The authors probably meant that for different λ s in one or more cities and for a common τ , then v is proportional to the explorable area. The text needs to be changed accordingly.

(3) L. 235-236: $A(\tau)$ for a typical daily τ is already a better measure than v from eq. 3 of “the

extension of the area that it is possible to explore in a typical working day". Meanwhile, v is the average speed at which an individual should expect to travel on their typical daily trips. Why is the first interpretation chosen rather than the second?

(4) Eq. 3: I am guessing from the supplementary information and from the legend of fig. 2 that "N is a normalisation constant" means that its role is to ensure that $\int f(t) dt = 1$ and this would explain why the division by $\int f(t) dt$ has been omitted in eq. 3. This omission should be properly indicated, otherwise the dimension of v is not $[m].[s]^{-1}$, but only $[m]$ and the integral becomes a sum rather than an average. More details about $f(t)$ and at the very least a reference to the supplementary material should appear in the main text. Also, presenting $f(t)$ as the direct result of a survey is misleading, since it is instead the result of some modelling work using an external methodology (which comes with its limits, see (6)).

(5) The description of $f(\tau)$ in the supplementary material is still hard to understand. What is the value and role of N? What is the quality of the fitting (e.g. R^2)? Why does T_{bus} mention only a bus when the survey is now about "transport habits" compared to "Oyster card journeys on bus, Tube, DLR and London Overground"?

(6) The study on which the $f(\tau)$ distribution is based only uses (significantly outdated) data from the UK. As a matter of fact, the authors of the original study lengthily discuss the limitations of their data. The only improvement made is a comparison to some more recent data that already shows discrepancies despite being from the same geographical region (which is not even featured in the case studies). This is an important flaw that should be mentioned in the discussion in the main text of the manuscript.

(7) L. 236-237: It would be useful to indicate that people are counted multiple times. For example, a typical trip from inside inner Paris is likely to stay inside inner Paris which only has a population of 2M, while fig. 2 suggests a typical traveller would reach up to 3M people in a day. On a side note, I would replace "meet" by "reach" on l. 237.

(8) L. 300: "without lack of generality" is a strong claim considering that this subset of cities contains only two clusters of very similar cities.

(9) L. 315-320: Please discuss the varying cultural expectations regarding car usage. Inequality in accessibility through public transport does not necessarily mean inequality in practical accessibility if the use of cars has been assumed a priori by urban planners.

(10) L. 341-354: This result is an artefact due to the combination of some hexagons being directly on the fastest lines together with a daily trip distribution that only represents the "average habit" over the entire city. People who are more "travel averse" will tend to pay a higher rent to be closer to the fastest lines, specifically to limit their "travel energy budget", without necessarily belonging to the 1% richest. An interesting analysis could have consisted in linking the hexagons with the social characteristics of their inhabitants and checking if the most socially privileged are also the most privileged in accessibility. Failing to do that, at least discuss the limits of this "1% of the hexagons" approach.

(11) L. 78-81 & l. 404-406: These statements are very subjective. It is a normal part of science to study a notion in all its aspects. As a matter of fact, a broader perspective on the subject would have been beneficial to the quality of the "Inequalities in urban accessibility patterns" section of the manuscript, as illustrated in the two previous points. Please rephrase (l. 78-81 in particular) in a more neutral and more moderate way.

Decision letter (RSOS-181315.R0)

29-Jan-2019

Dear Dr Biazzo,

The editors assigned to your paper ("Universal scores for accessibility and inequalities in urban areas") have now received comments from reviewers. We would like you to revise your paper in accordance with the referee and Associate Editor suggestions which can be found below (not including confidential reports to the Editor). Please note this decision does not guarantee eventual acceptance.

Please submit a copy of your revised paper before 21-Feb-2019. Please note that the revision deadline will expire at 00.00am on this date. If we do not hear from you within this time then it will be assumed that the paper has been withdrawn. In exceptional circumstances, extensions may be possible if agreed with the Editorial Office in advance. We do not allow multiple rounds of revision so we urge you to make every effort to fully address all of the comments at this stage. If deemed necessary by the Editors, your manuscript will be sent back to one or more of the original reviewers for assessment. If the original reviewers are not available, we may invite new reviewers.

- Data accessibility

If you wish to submit your supporting data or code to Dryad (<http://datadryad.org/>), or modify your current submission to dryad, please use the following link:
<http://datadryad.org/submit?journalID=RSOS&manu=RSOS-181315>

- **Competing interests**

- **Authors' contributions**

- **Acknowledgements**

- **Funding statement**

Kind regards,
Andrew Dunn
Senior Publishing Editor
Royal Society Open Science Editorial Office
Royal Society Open Science
openscience@royalsociety.org

on behalf of Prof Miles Padgett (Subject Editor)
openscience@royalsociety.org

Comments to Author:

Reviewers' Comments to Author:

Reviewer: 1

Comments to the Author(s)

Overall, I think this is a very interesting paper. This paper makes a unique contribution in offering a new method in measuring temporal accessibility and its relevance with social equity. The platform created provides excellent examples of visualizations of temporal accessibility in several world cities and has potential implications for transport policies, particularly in transport equity. Other cities may easily create similar accessibility maps using the codes provided by the authors.

Reviewer: 2

Comments to the Author(s)

One of the interesting things about the “open review” process at the Royal Society Open Science journal is the idea of an ‘unmasked’ peer review: where I can explicate my own academic personality as a background to my reviewer judgment. So to start with that, I am Dr. Michiel van Meeteren, lecturer in human geography at Loughborough University. The reason why I believe I was chosen as a reviewer to this paper is because I was a co-author on the following paper:

“Boussauw, K., Van Meeteren, M., Sansen, J., Meijers, E., Storme, T., Louw, E., Derudder, B. & Witlox, F. (2018). Planning for agglomeration economies in a polycentric region: Envisioning an efficient metropolitan core area in Flanders, *European Journal of Spatial Development*, 69.<http://doi.org/10.30689/EJSD2018:69.1650-9544>”

In this paper we combine accessibility analysis with isochrone maps for a practical application in public transport planning. In doing so we explicitly draw on the long established history of using these techniques in transport geography and related disciplines. I have to admit up front that the method proposed in this paper is more sophisticated as what we used, and I am actually quite enthusiastic about the indicators that this paper proposes. However, my disciplinary background is also a source of criticism in this review, in the sense that “the claims to newness” in this paper does disenfranchise the contributions from geography and geographers to the study of accessibility somewhat. Claims that this paper is “a first contribution in this direction” (abstract), that “studying cities in terms of travel time is a minoritarian view” (page 2, for geographers it is the default view). “very few people have studied the temporal features of these systems” (page 2). tie into this feeling. It might be new to complexity science, computer science or physics, but it is not new to geographers at all. This is something the authors acknowledge when they cite some of geography’s foundational works, references 21-29 contain some of the classic papers I would normally recommend. This resonates with a broader annoyance as data science, physics and complexity theorists have been ‘discovering the city’ as a topic of research. We as geographers welcome new perspectives and input to some of our established research topics, but we do like to have some acknowledgement of our historical contribution to a topic, a recurring complaint nicely elaborated in O’ Sullivan and Manson (2015).

The authors mention that “that the science of cities is a relatively new research area (P.2)”. I would make the case that it needs qualification. I’d say that the author of that claim, Michael Batty has been doing the science of cities for the whole of his long and distinguished career and that he himself was able to establish that career on a network of scholars in the 1960s (Barnes and Wilson 2014 provide historical perspective; and I did my little bit in Van Meeteren and Poorthuis 2018). In fact, accessibility analysis goes back to the collaboration of a geographer and a physicist on population potential in the 1950s (Stewart and Warntz, 1958). In other words, this reviewer would be very happy to see this contribution not as the “invention” of a debate/concept but as a “contribution/refinement” to a longstanding, but perhaps at certain times in history dormant research agenda. Most of this revision is just a question of language and due citation. Even

nowadays there is quite a bustling community of geographers working on these topics, for instance just look at journals such as “Environment and planning B: urban analytics and city science” (edited by Mike Batty), *The Journal of Transport Geography* or *Transportation Research*. So although I agree with the authors that big data and massive computation offers great potential to revisit these topics (page 4), it is not that there is not a large community of scholars doing exactly that.

Which brings me to another comment: I don’t think this community of geographers has no standard measures of accessibility: we have (see Boussauw et al 2018 for references) and they are quite similar in type to what this paper proposes.

That said, I am very enthusiastic about the new measures “velocity score”, “sociality score” and “city sociality” and I like the crisp and unambiguous way in which the authors define it. Those measures will be a very welcome and useful tool for comparative research across cities both for academic and for practical applications. Geographers do not like describing indicators with adjectives like “universal” or “law” for all sorts of epistemological and historical reasons (Barnes 2013). Nevertheless I understand there is a problem of translation between disciplinary languages here.

To conclude there is one final weaker point in the paper I would like to point out. The section “Space-time distribution of inequality in accessibility patterns” (page 9-10) assumes a monocentric model of a city. This monocentric model has been increasingly found to be unrealistic of city structures regardless of where you are (Van Meeteren et al. 2016 is a good introduction to these debates see also Clark and Kuijpers-Line 1994, Clark 2000). Of course there are car based and public transport based cities and North American cities tend to be the former for historical reasons. So the rankings provided do not surprise a geographer at all. They are exactly what one would expect based on geographical knowledge. However, if the paper is serious in its ambition to provide a cartographic tool that helps policy makers, it should take into account that cities have several employment centres and that the traditional ‘urban core’ has long since lost its meaning as the principal centre of employment public transport is directed toward. Most new jobs come in the city’s edge (Phelps, 2017; Keil, 2017), therefore championing cities that have the best accessibility to the city center is not always the best policy advice, although advocating for higher densities that allow for public transportation surely is. On that note, there is a literature on ‘threshold values’ when public transport becomes viable. This is not so much a “general mechanism” (page 10) as the simple result of transport economics that require a minimal number of customers for an investment to be politically and economically sustainable. Again here we encounter some disciplinary loss in translation. For social scientists these “general mechanism” are the product of social and political decisions which makes us cautious in using terms like “general” and “universal” as society could easily change that reality if enough people want to.

Lastly, the ‘most of the world now lives in cities’ argument is now so much of a cliché (that hides a lot, see Brenner and Schmid 2014) that I would avoid using it.

I hope these points allow the author to improve on what is already a really interesting paper. I look forward to see it develop further.

Barnes, T. J. (2013). Big data, little history. *Dialogues in Human Geography*, 3(3), 297–302.

Barnes, T. J., & Wilson, M. W. (2014). Big Data, social physics, and spatial analysis: The early years. *Big Data & Society*, 1(1), 1–14.

Boussauw, K., Van Meeteren, M., Sansen, J., Meijers, E., Storme, T., Louw, E., Derudder, B. &

Witlox, F. (2018). Planning for agglomeration economies in a polycentric region: Envisioning an efficient metropolitan core area in Flanders, *European Journal of Spatial Development*, 69. <http://doi.org/10.30689/EJSD2018:69.1650-9544>

Brenner, N., & Schmid, C. (2014). The “Urban Age” in Question. *International Journal of Urban and Regional Research*, 38(3), 731–755.

Clark, W. A. V., & Kuijpers-Linde, M. (1994). Commuting in restructuring urban regions. *Urban Studies*, 31(3), 465–483.

Clark, W. A. V. (2000). Monocentric to Policentric: New Urban Forms and Old Paradigms. In G. Bridge & S. Watson (Eds.), *A Companion to the City* (pp. 141–155). Malden/Oxford: Blackwell Publishing.

Keil, R. (2017). *Suburban planet*. John Wiley & Sons

O’Sullivan, D., & Manson, S. M. (2015). Do Physicists Have Geography Envy? And What Can Geographers Learn from It?. *Annals of the Association of American geographers*, 105(4), 704–722.

Phelps, N. (2017). *Interplaces: An Economic Geography of the Inter-urban and International Economies*. Oxford University Press

Stewart, J. Q., & Warntz, W. (1958). Macrogeography and Social Science. *Geographical Review*, 48(2), 167.

Van Meeteren, M., Poorthuis, A., Derudder, B., & Witlox, F. (2016). Pacifying Babel’s Tower: A scientometric analysis of polycentricity in urban research. *Urban Studies*, 53(6), 1278–1298. <http://doi.org/10.1177/0042098015573455>

Van Meeteren, M., & Poorthuis, A. (2018). Christaller and “big data”: recalibrating central place theory via the geoweb. *Urban Geography*, 39(1), 122–148.

Reviewer: 3

Comments to the Author(s)

If my assumptions are correct, then the methodology is sound. However, the use of imprecise scientific terminology and the lack of rigour when explaining the mathematical equations often forces the reader to guess what the authors meant instead of what they wrote. This becomes particularly problematic when it results in masking some important limitations of the methodology. Although I believe that the following points should prove relatively easy to fix, I would strongly advise against publishing the manuscript until these are addressed, hence the major revision recommendation.

(1) L. 206: The speed of expansion of the isochrone is $dr/d\tau$, not r/τ . Please indicate the correct interpretation (l. 208–209) directly. In addition, the term “velocity” is generally used instead of “speed” when the direction is considered, while it has precisely been removed here.

(2) L. 209–210: v is certainly not proportional to the explorable area from the hexagon λ . If an isochrone moves at a constant speed of 1 m.s⁻¹ and is a perfect circle, then $v(1)=1$ and $A(1)=\pi$, while $v(2)=1$ and $A(2)=4\pi$, and more generally, $v(n)=1$ and $A(n)=n^2\pi$ for any n . The authors probably meant that for different λ s in one or more cities and for a common τ , then v is proportional to the explorable area. The text needs to be changed accordingly.

- (3) L. 235-236: $A(\tau)$ for a typical daily τ is already a better measure than v from eq. 3 of “the extension of the area that it is possible to explore in a typical working day”. Meanwhile, v is the average speed at which an individual should expect to travel on their typical daily trips. Why is the first interpretation chosen rather than the second?
- (4) Eq. 3: I am guessing from the supplementary information and from the legend of fig. 2 that “ N is a normalisation constant” means that its role is to ensure that $\int f(t) dt = 1$ and this would explain why the division by $\int f(t) dt$ has been omitted in eq. 3. This omission should be properly indicated, otherwise the dimension of v is not $[m].[s]^{-1}$, but only $[m]$ and the integral becomes a sum rather than an average. More details about $f(t)$ and at the very least a reference to the supplementary material should appear in the main text. Also, presenting $f(t)$ as the direct result of a survey is misleading, since it is instead the result of some modelling work using an external methodology (which comes with its limits, see (6)).
- (5) The description of $f(\tau)$ in the supplementary material is still hard to understand. What is the value and role of N ? What is the quality of the fitting (e.g. R^2)? Why does T_{bus} mention only a bus when the survey is now about “transport habits” compared to “Oyster card journeys on bus, Tube, DLR and London Overground”?
- (6) The study on which the $f(\tau)$ distribution is based only uses (significantly outdated) data from the UK. As a matter of fact, the authors of the original study lengthily discuss the limitations of their data. The only improvement made is a comparison to some more recent data that already shows discrepancies despite being from the same geographical region (which is not even featured in the case studies). This is an important flaw that should be mentioned in the discussion in the main text of the manuscript.
- (7) L. 236-237: It would be useful to indicate that people are counted multiple times. For example, a typical trip from inside inner Paris is likely to stay inside inner Paris which only has a population of 2M, while fig. 2 suggests a typical traveller would reach up to 3M people in a day. On a side note, I would replace “meet” by “reach” on l. 237.
- (8) L. 300: “without lack of generality” is a strong claim considering that this subset of cities contains only two clusters of very similar cities.
- (9) L. 315-320: Please discuss the varying cultural expectations regarding car usage. Inequality in accessibility through public transport does not necessarily mean inequality in practical accessibility if the use of cars has been assumed a priori by urban planners.
- (10) L. 341-354: This result is an artefact due to the combination of some hexagons being directly on the fastest lines together with a daily trip distribution that only represents the “average habit” over the entire city. People who are more “travel averse” will tend to pay a higher rent to be closer to the fastest lines, specifically to limit their “travel energy budget”, without necessarily belonging to the 1% richest. An interesting analysis could have consisted in linking the hexagons with the social characteristics of their inhabitants and checking if the most socially privileged are also the most privileged in accessibility. Failing to do that, at least discuss the limits of this “1% of the hexagons” approach.
- (11) L. 78-81 & l. 404-406: These statements are very subjective. It is a normal part of science to study a notion in all its aspects. As a matter of fact, a broader perspective on the subject would have been beneficial to the quality of the “Inequalities in urban accessibility patterns” section of the manuscript, as illustrated in the two previous points. Please rephrase (l. 78-81 in particular) in a more neutral and more moderate way.

Author's Response to Decision Letter for (RSOS-181315.R0)

See Appendix A.

RSOS-181315.R1 (Revision)

Review form: Reviewer 2 (Michiel Van Meeteren)

Is the manuscript scientifically sound in its present form?

Yes

Are the interpretations and conclusions justified by the results?

Yes

Is the language acceptable?

No

Is it clear how to access all supporting data?

Yes

Do you have any ethical concerns with this paper?

No

Have you any concerns about statistical analyses in this paper?

I do not feel qualified to assess the statistics

Recommendation?

Accept with minor revision (please list in comments)

Comments to the Author(s)

I now had the chance to review the revision of Universal scores for accessibility in urban areas. The major concerns I had with the previous version have sufficiently been resolved. What remains are some minor issues, mostly of a semantic nature, that can be easily resolved.

1: One overarching comment is that the paper would benefit from a thorough language edit. Although everything is understandable, there are repeated minor grammatical and spelling errors and many dangling modifiers (where the reference object in the previous sentence is unclear), and repeated language that could be avoided by use of synonyms. A language specialist might choose different adjectives / phrases in order to get a more nuanced perspective across, and sentences could be split to improve readability. For instance the first sentence of the introduction is more than four lines, which is not very common in the English language (contrary to many Latin languages). "Lost in translation" issues could also be in play in some of the other minor things I raise below.

2: What do you mean with "the fruition" of metrics? (page 12, line 32). I agree that your metrics are intuitive, but do you mean "making their practical application easier" or something like that?

3: Both in the abstract and on pages 18-19 there is the claim that there is a "general mechanism" or "common mechanism" at work that can possibly explain the regular patterns of inequality found. I would advise some caution here. The notion of "mechanism" suggests ideas about causation of spatial inequality which is somewhat outside the scope of the current paper. I wholeheartedly agree that there is a recurring pattern across cities, and that this seemingly "universal pattern" merits further investigation. However, what causes these patterns is a question that is yet to be answered. Even if one was able to describe the mechanism in a morphological sense (for instance through a process of preferential attachment) it still could have different causes. Some may be described in the realm of transport economics (minimum densities to profitably exploit public transport or alternatively through network externalities), labour market geographies (places of work and places of living), theories of rent and so on. I would suggest being slightly more modest and claim that there is a recurring morphological pattern of inequality in accessibility across cities that is very interesting to disentangle in future urban-scientific research. But like my similar remark in the previous review round: I am aware that this is viewed differently across disciplines, and that where I come from we tend to be very cautious when it concerns causality.

4: Related, I have still some issue with the generous use of the term "universal" (with all its "law-like" connotations in epistemology). Although the term is qualified as meaning "in the sense that they can be applied in every city and in different context allowing comparison between different areas and means of transportation." (page 12, line 67), in my world we would not use the term universal for that condition. Perhaps "universally applicable", "Generic indicators" or "context-independent measures/indicators" would be my preferred terms. I have a similar feeling with the paper's title "universal scores for accessibility and inequalities in urban areas". "Universally applicable accessibility indicators and inequalities in urban areas" would be closer to my preferred title for what the paper is about. A "universal score" seems too timeless, too much cast in stone for a social object such as the city. But again, these things are the "lost in translations" common to interdisciplinary dialogue.

With all those semantic remarks I might almost overlook to remark how much I consider the proposed indicators a valuable addition to the toolbox of researchers working in this field, and that I enjoyed getting acquainted with them and would surely consider applying them in future research. Thus I still warmly recommend this paper for eventual publication, perhaps after giving some thoughts to my final semantic concerns.

Review form: Reviewer 3

Is the manuscript scientifically sound in its present form?

Yes

Are the interpretations and conclusions justified by the results?

No

Is the language acceptable?

Yes

Is it clear how to access all supporting data?

Yes

Do you have any ethical concerns with this paper?

No

Have you any concerns about statistical analyses in this paper?

No

Recommendation?

Major revision is needed (please make suggestions in comments)

Comments to the Author(s)

I am only partly satisfied with the corrections made by the authors. Point (2) below, for example, remains a major issue for me. I do understand that the authors made the most of what was available to them, however I fail to understand why it is not possible for them to acknowledge properly in the manuscript serious limitations to the validity of their results. Claiming that the Supplementary Material ascertains the robustness of the methodology is misleading, if not dishonest.

In addition, the authors make bold claims of providing “operational” information that policy makers were in dire need of, but seem opposed to including in the discussion any argument that would explain why their results are so disconnected from the reality of the field (points (3)-(5)).

(1) We have the feeling that, when speaking about transportations, is easier for all to quantify and imagine a speed rather than an area as a function of time, so it is easier to make a comparison with our everyday life and communicate the scientific results to a large audience with very different backgrounds.

I agree. Hence, I don't understand why the first clear and neatly written interpretation of $v(\lambda)$ (given l. 238), the one people are likely to remember, is “the Velocity Score provides with a measure of the extension of the area” rather than the speed interpretation.

(2) We used empirical travel-time distributions that we were able to find in literature. We could find only two analytic forms derived from empirical data in the literature. Then we add a third one, the “flat” distribution, just to check if our results are sensitive to the choice of the travel time distributions. What the Fig.1 of the Supporting Material file S1 shows is indeed that our results are robust against a different choice of travel-time distributions.

Comparing two travel time distributions obtained for one city, then applying these distributions to completely different cities is the exact opposite of a “robust” approach. Where is it justified in the manuscript that the empirical daily budget for London residents is relevant to Boston or Sydney? Why should we expect people living in Florence or Toulouse to devote as much time to transport as people living in a city 15 times bigger? In addition, what is the quality of these fittings compared to real data, not compared to other modelled data?

(3) In our approach, people are not counted multiple times. We use the definition of an urban area as economic functional units given by OECD/EU, as written in the main text, in the Methods section. In the case of Paris, the urban area contains more than 12 millions of people. Due to the high level of development of the public transports, from the center of Paris, one can reach more than 3 millions of people, some of them, of course, living outside the municipality of Paris, though still within its urban area.

The methodology assumes that the daily budget is entirely spent in one round trip. If, for example, an individual undertakes a first trip from inside Paris to their workplace 4 km away and a second trip to a shop 2 km away, their movements remain in fact bounded within a disk of radius 4 km, representing 64% of Paris, or roughly 1.3M inhabitants. However, the method would count them as traveling up to 6km away, reaching roughly a total of 3M inhabitants. In this case,

the 320k inhabitants of the area of the second trip, already counted once in the first trip, add an extra 1.7M to the total count of people reachable. This needs to be properly indicated.

There are already one unrealistic isotropy assumption (contradicted, for example, by the lack of willingness to cross the “périphérique” - or purpose to do so - for inhabitants of the municipality) and one incorrect assumption that the daily budget is uniformly distributed inside the city (people living inside the “périphérique” do so at a very high cost to avoid spending time in transports). As a result, the sociality score obtained for inner Paris residents is completely unrealistic. In fact, the maps in panel B of figure 2 appear to be a less precise and less accurate version of this map, which provides simple objective information without making any claim of representing anything “typical”:

<http://www.urbanmorphologyinstitute.org/france/>

(4) In the section “Inequalities in urban accessibility patterns” we study the inequalities in the access to the public transports among populations, not the absolute values. The point is that we observe common patterns among cities regardless if the urban planners design the city to be more cars or public transports oriented.

The purpose of the manuscript is to propose a “unifying definition of accessibility” that would be directly useful to policy makers. Ignoring how urban planners design their cities does not sound like a good starting point to achieve that. I fail to see why it is not possible to add one or two lines pointing out the role of cars as an explanation to the difference in the observed results between cities.

(5) Our analysis is trying to measure the performance of public transports to cover the space and connect people regardless of the “habit” of the people. As previously explained, our results are quite stable against a reasonable choice of the “average habit” choose. We do not consider the 1% richest and the rest 99%. We only consider people or area with accessibility 1% higher in comparison to the rest 99%. The analysis of the connections with the economic, cultural or habits diversity will be the subject of future studies.

The “1% highest accessibility hexagons” is not necessarily meaningful. An individual living in the south of Horley (42 km from London) will fare very well since they will be in the same hexagon as the Gatwick express (only £18/trip) and the fast suburban Southern and Thameslink trains. Meanwhile an individual living on Westbourne Grove in zone 1 may (depending on the hexagon placement) have to rely on buses or walk to another hexagon to take the tube.

By construction, building a transport network requires to fill a 2-dimensional space with 1-dimensional lines. As cities become larger, this is less and less possible, so planners only try to connect people to places that matter to them. They try to cater to people’s habits, no matter the number of random strangers they can wave hello to from the window of their trains as they pass by towards their intended destination. This is similar to the difference between degree centrality and Katz centrality in network theory. What matters most: the number of nodes one is connected to or the importance of the nodes one is connected to?

Some may think that taking these considerations into account participates in the “proliferation of definitions that disperse the scientific effort”. However, the “top 1%” analysis presented in the manuscript is misleading and does not add any valuable “operational” information. It does not reveal anything about inequality in “access to opportunities”, only that central inhabitants could easily go check if the birds are different in the suburban residential areas if they wanted to and that airports and amusement parks are the best spots for high accessibility unauthorized camping.

Decision letter (RSOS-181315.R1)

24-Apr-2019

Dear Dr Biazzo:

I write you in regards to manuscript # RSOS-181315.R1 entitled "Universal scores for accessibility and inequalities in urban areas" which you submitted to Royal Society Open Science.

Regrettably, in view of the criticisms of the reviewer(s) found at the bottom of this letter, your manuscript has been denied publication in Royal Society Open Science.

Thank you for considering Royal Society Open Science for the publication of your research. I hope the outcome of this specific submission will not discourage you from the submission of future manuscripts.

on behalf of Prof Miles Padgett (Subject Editor)
openscience@royalsociety.org

Associate Editor Comments to Author:

While both reviewers consider the paper to have made improvements, one of the reviewers remains critical of the manuscript, and your responses to their concerns. Regrettably, Royal Society Open Science does not generally allow multiple rounds of revision to be conducted if the reviewers are not satisfied by the changes incorporated. With this in mind, we are unfortunately not able to consider your manuscript further for publication. We hope the reviewers' comments are useful if you choose to submit the manuscript for consideration elsewhere. Thanks for considering the journal on this occasion.

Reviewer comments to Author:

Reviewer: 3

Comments to the Author(s)

I am only partly satisfied with the corrections made by the authors. Point (2) below, for example, remains a major issue for me. I do understand that the authors made the most of what was available to them, however I fail to understand why it is not possible for them to acknowledge properly in the manuscript serious limitations to the validity of their results. Claiming that the Supplementary Material ascertains the robustness of the methodology is misleading, if not dishonest.

In addition, the authors make bold claims of providing "operational" information that policy makers were in dire need of, but seem opposed to including in the discussion any argument that would explain why their results are so disconnected from the reality of the field (points (3)-(5)).

(1) We have the feeling that, when speaking about transportations, is easier for all to quantify and imagine a speed rather than an area as a function of time, so it is easier to make a comparison with our everyday life and communicate the scientific results to a large audience with very different backgrounds.

I agree. Hence, I don't understand why the first clear and neatly written interpretation of $v(?)$ (given l. 238), the one people are likely to remember, is "the Velocity Score provides with a measure of the extension of the area" rather than the speed interpretation.

(2) We used empirical travel-time distributions that we were able to find in literature. We could find only two analytic forms derived from empirical data in the literature. Then we add a third one, the "flat" distribution, just to check if our results are sensitive to the choice of the travel time distributions. What the Fig.1 of the Supporting Material file S1 shows is indeed that our results are robust against a different choice of travel-time distributions.

Comparing two travel time distributions obtained for one city, then applying these distributions to completely different cities is the exact opposite of a "robust" approach. Where is it justified in the manuscript that the empirical daily budget for London residents is relevant to Boston or Sydney? Why should we expect people living in Florence or Toulouse to devote as much time to transport as people living in a city 15 times bigger? In addition, what is the quality of these fittings compared to real data, not compared to other modelled data?

(3) In our approach, people are not counted multiple times. We use the definition of an urban area as economic functional units given by OECD/EU, as written in the main text, in the Methods section. In the case of Paris, the urban area contains more than 12 millions of people. Due to the high level of development of the public transports, from the center of Paris, one can reach more than 3 millions of people, some of them, of course, living outside the municipality of Paris, though still within its urban area.

The methodology assumes that the daily budget is entirely spent in one round trip. If, for example, an individual undertakes a first trip from inside Paris to their workplace 4 km away and a second trip to a shop 2 km away, their movements remain in fact bounded within a disk of radius 4 km, representing 64% of Paris, or roughly 1.3M inhabitants. However, the method would count them as traveling up to 6km away, reaching roughly a total of 3M inhabitants. In this case, the 320k inhabitants of the area of the second trip, already counted once in the first trip, add an extra 1.7M to the total count of people reachable. This needs to be properly indicated.

There are already one unrealistic isotropy assumption (contradicted, for example, by the lack of willingness to cross the "périphérique" - or purpose to do so - for inhabitants of the municipality) and one incorrect assumption that the daily budget is uniformly distributed inside the city (people living inside the "périphérique" do so at a very high cost to avoid spending time in transports). As a result, the sociality score obtained for inner Paris residents is completely unrealistic. In fact, the maps in panel B of figure 2 appear to be a less precise and less accurate version of this map, which provides simple objective information without making any claim of representing anything "typical":

<http://www.urbanmorphologyinstitute.org/france/>

(4) In the section "Inequalities in urban accessibility patterns" we study the inequalities in the access to the public transports among populations, not the absolute values. The point is that we observe common patterns among cities regardless if the urban planners design the city to be more cars or public transports oriented.

The purpose of the manuscript is to propose a “unifying definition of accessibility” that would be directly useful to policy makers. Ignoring how urban planners design their cities does not sound like a good starting point to achieve that. I fail to see why it is not possible to add one or two lines pointing out the role of cars as an explanation to the difference in the observed results between cities.

(5) Our analysis is trying to measure the performance of public transports to cover the space and connect people regardless of the “habit” of the people. As previously explained, our results are quite stable against a reasonable choice of the “average habit” choose. We do not consider the 1% richest and the rest 99%. We only consider people or area with accessibility 1% higher in comparison to the rest 99%. The analysis of the connections with the economic, cultural or habits diversity will be the subject of future studies.

The “1% highest accessibility hexagons” is not necessarily meaningful. An individual living in the south of Horley (42 km from London) will fare very well since they will be in the same hexagon as the Gatwick express (only £18/trip) and the fast suburban Southern and Thameslink trains. Meanwhile an individual living on Westbourne Grove in zone 1 may (depending on the hexagon placement) have to rely on buses or walk to another hexagon to take the tube.

By construction, building a transport network requires to fill a 2-dimensional space with 1-dimensional lines. As cities become larger, this is less and less possible, so planners only try to connect people to places that matter to them. They try to cater to people’s habits, no matter the number of random strangers they can wave hello to from the window of their trains as they pass by towards their intended destination. This is similar to the difference between degree centrality and Katz centrality in network theory. What matters most: the number of nodes one is connected to or the importance of the nodes one is connected to?

Some may think that taking these considerations into account participates in the “proliferation of definitions that disperse the scientific effort”. However, the “top 1%” analysis presented in the manuscript is misleading and does not add any valuable “operational” information. It does not reveal anything about inequality in “access to opportunities”, only that central inhabitants could easily go check if the birds are different in the suburban residential areas if they wanted to and that airports and amusement parks are the best spots for high accessibility unauthorized camping.

Reviewer: 2

Comments to the Author(s)

I now had the chance to review the revision of Universal scores for accessibility in urban areas. The major concerns I had with the previous version have sufficiently been resolved. What remains are some minor issues, mostly of a semantic nature, that can be easily resolved.

1: One overarching comment is that the paper would benefit from a thorough language edit. Although everything is understandable, there are repeated minor grammatical and spelling errors and many dangling modifiers (where the reference object in the previous sentence is unclear), and repeated language that could be avoided by use of synonyms. A language specialist might choose different adjectives / phrases in order to get a more nuanced perspective across, and sentences could be split to improve readability. For instance the first sentence of the introduction is more than four lines, which is not very common in the English language (contrary to many Latin languages). "Lost in translation" issues could also be in play in some of the other minor things I raise below.

2: What do you mean with "the fruition" of metrics? (page 12, line 32). I agree that your metrics are intuitive, but do you mean "making their practical application easier" or something like that?

3: Both in the abstract and on pages 18-19 there is the claim that there is a "general mechanism" or "common mechanism" at work that can possibly explain the regular patterns of inequality found. I would advise some caution here. The notion of "mechanism" suggests ideas about causation of spatial inequality which is somewhat outside the scope of the current paper. I wholeheartedly agree that there is a recurring pattern across cities, and that this seemingly "universal pattern" merits further investigation. However, what causes these patterns is a question that is yet to be answered. Even if one was able to describe the mechanism in a morphological sense (for instance through a process of preferential attachment) it still could have different causes. Some may be described in the realm of transport economics (minimum densities to profitably exploit public transport or alternatively through network externalities), labour market geographies (places of work and places of living), theories of rent and so on. I would suggest being slightly more modest and claim that there is a recurring morphological pattern of inequality in accessibility across cities that is very interesting to disentangle in future urban-scientific research. But like my similar remark in the previous review round: I am aware that this is viewed differently across disciplines, and that where I come from we tend to be very cautious when it concerns causality.

4: Related, I have still some issue with the generous use of the term "universal" (with all its "law-like" connotations in epistemology). Although the term is qualified as meaning "in the sense that they can be applied in every city and in different context allowing comparison between different areas and means of transportation." (page 12, line 67), in my world we would not use the term universal for that condition. Perhaps "universally applicable", "Generic indicators" or "context-independent measures/indicators" would be my preferred terms. I have a similar feeling with the paper's title "universal scores for accessibility and inequalities in urban areas". "Universally applicable accessibility indicators and inequalities in urban areas" would be closer to my preferred title for what the paper is about. A "universal score" seems too timeless, too much cast in stone for a social object such as the city. But again, these things are the "lost in translations" common to interdisciplinary dialogue.

With all those semantic remarks I might almost overlook to remark how much I consider the proposed indicators a valuable addition to the toolbox of researchers working in this field, and that I enjoyed getting acquainted with them and would surely consider applying them in future research. Thus I still warmly recommend this paper for eventual publication, perhaps after giving some thoughts to my final semantic concerns.

Author's Response to Decision Letter for (RSOS-181315.R1)

See Appendix B.

RSOS-190979.R0

Review form: Reviewer 4

Is the manuscript scientifically sound in its present form?

Yes

Are the interpretations and conclusions justified by the results?

No

Is the language acceptable?

No

Do you have any ethical concerns with this paper?

No

Have you any concerns about statistical analyses in this paper?

No

Recommendation?

Accept with minor revision (please list in comments)

Comments to the Author(s)

I believe the paper does make a contribution to urban accessibility studies by proposing indices that can be used to compare pedestrian and public transit accessibility levels from city to city. I think the paper deserves publication, but there are still a few issues the paper should address before publication, however.

First, as has already been indicated in previews reviews, a thorough language edit is still needed by a native English speaker.

Second, the most important issue I find with the paper is the assumption that trips are modeled from all hexagons to all hexagons. I understand the simple appeal of this assumption – which is also common in computer science and graph theory algorithms, where indices are computed from all nodes to all nodes. It offers a simple way to make cities comparable to each other, but it doesn't really describe real commuting flows nor aims of transit systems.

Urban populations do not commute with equal likelihood to all parts of the city. Trips are much more common from homes to jobs, homes to commercial destinations as well as homes to recreational destinations and vica versa These destinations types are never uniformly distributed in space – they tend to concentrate in the CBD and other subcenters. Accordingly, public transportation systems and street networks evolve over time to prioritize access to such destinations instead of maximizing uniform access to all parts of the city. If we measured how easily people from all parts of the city can gain access to jobs as well as commercial destinations, we would likely find different results. The 1% VS 99% access differences highlighted in the paper would likely change. And the meaning of velocity, sociality and cohesion would likely change. People meet others not at their homes, but in public or commercial and recreational spaces of the city. We should thus not judge the quality of public transit systems based on how well they allow residents to get to all other residents' homes.

To address this issue, I suggest that:

- a) the discussion section should either eliminate or substantially change the discussion about the inequality of access in cities.
- b) Some material should be added towards the beginning of the paper about the all hexagons to all hexagons O-D choice.
- c) The discussion section should also mention the need in future work to test a similar approach with different assumptions about origins and destinations.

d) The paper should reduce its claims that an end-all approach is delivered to compare cities on accessibility levels. Rather, the paper makes an interesting step towards comparable urban accessibility indices but there is a quite a bit more to do before a universal or standardized approach can be agreed upon.

Decision letter (RSOS-190979.R0)

23-Jul-2019

Dear Dr Biazzo

On behalf of the Editor, I am pleased to inform you that your Manuscript RSOS-190979 entitled "General scores for accessibility and inequality measures in urban areas" has been accepted for publication in Royal Society Open Science subject to minor revision in accordance with the referee suggestions. Please find the referees' comments at the end of this email.

The reviewers and Subject Editor have recommended publication, but also suggest some minor revisions to your manuscript. Therefore, I invite you to respond to the comments and revise your manuscript.

- Ethics statement

- Data accessibility

<http://datadryad.org/submit?journalID=RSOS&manu=RSOS-190979>

- Competing interests

- Authors' contributions

All submissions, other than those with a single author, must include an Authors' Contributions section which individually lists the specific contribution of each author. The list of Authors should meet all of the following criteria; 1) substantial contributions to conception and design, or

acquisition of data, or analysis and interpretation of data; 2) drafting the article or revising it critically for important intellectual content; and 3) final approval of the version to be published.

- Acknowledgements

- Funding statement

Because the schedule for publication is very tight, it is a condition of publication that you submit the revised version of your manuscript before 01-Aug-2019. Please note that the revision deadline will expire at 00.00am on this date. If you do not think you will be able to meet this date please let me know immediately.

- 1) A text file of the manuscript (tex, txt, rtf, docx or doc), references, tables (including captions) and figure captions. Do not upload a PDF as your "Main Document".
- 2) A separate electronic file of each figure (EPS or print-quality PDF preferred (either format should be produced directly from original creation package), or original software format)
- 3) Included a 100 word media summary of your paper when requested at submission. Please ensure you have entered correct contact details (email, institution and telephone) in your user account

4) Included the raw data to support the claims made in your paper. You can either include your data as electronic supplementary material or upload to a repository and include the relevant doi within your manuscript

5) All supplementary materials accompanying an accepted article will be treated as in their final form. Note that the Royal Society will neither edit nor typeset supplementary material and it will be hosted as provided. Please ensure that the supplementary material includes the paper details where possible (authors, article title, journal name).

Kind regards,
Andrew Dunn
Senior Publishing Editor
Royal Society Open Science
openscience@royalsociety.org

on behalf of Prof Miles Padgett (Subject Editor)
openscience@royalsociety.org

Associate Editor Comments to Author:

Thank you for the resubmission. Given the degree of oversight earlier iterations of the paper have received, the Editors are prepared to make an assessment of this version on the basis of only one reviewer's report. This is a new reviewer, and you'll see that they are broadly in favour of accepting your paper; however, they raise a number of points you must respond to before we can accept the paper in its final form. In particular, as has been noted previously, you must seek further language polishing - examples of services offering this are available at <https://royalsociety.org/journals/authors/language-polishing/>.

Please ensure you provide evidence that language editing has been provided with your revision, as well as a point-by-point response and a marked-up version of the manuscript showing the changes you've made.

If you do not do as the reviewer has asked, we will not be able to consider the paper further.

Reviewer comments to Author:
Reviewer: 4

Comments to the Author(s)

I believe the paper does make a contribution to urban accessibility studies by proposing indices that can be used to compare pedestrian and public transit accessibility levels from city to city. I think the paper deserves publication, but there are still a few issues the paper should address before publication, however.

First, as has already been indicated in previews reviews, a thorough language edit is still needed by a native English speaker.

Second, the most important issue I find with the paper is the assumption that trips are modeled from all hexagons to all hexagons. I understand the simple appeal of this assumption – which is also common in computer science and graph theory algorithms, where indices are computed from all nodes to all nodes. It offers a simple way to make cities comparable to each other, but it doesn't really describe real commuting flows nor aims of transit systems.

Urban populations do not commute with equal likelihood to all parts of the city. Trips are much more common from homes to jobs, homes to commercial destinations as well as homes to recreational destinations and vica versa These destinations types are never uniformly distributed in space – they tend to concentrate in the CBD and other subcenters. Accordingly, public transportation systems and street networks evolve over time to prioritize access to such destinations instead of maximizing uniform access to all parts of the city. If we measured how easily people from all parts of the city can gain access to jobs as well as commercial destinations, we would likely find different results. The 1% VS 99% access differences highlighted in the paper would likely change. And the meaning of velocity, sociality and cohesion would likely change. People meet others not at their homes, but in public or commercial and recreational spaces of the city. We should thus not judge the quality of public transit systems based on how well they allow residents to get to all other residents' homes.

To address this issue, I suggest that:

- a) the discussion section should either eliminate or substantially change the discussion about the inequality of access in cities.
- b) Some material should be added towards the beginning of the paper about the all hexagons to all hexagons O-D choice.
- c) The discussion section should also mention the need in future work to test a similar approach with different assumptions about origins and destinations.
- d) The paper should reduce its claims that an end-all approach is delivered to compare cities on accessibility levels. Rather, the paper makes an interesting step towards comparable urban accessibility indices but there is a quite a bit more to do before a universal or standardized approach can be agreed upon.

Author's Response to Decision Letter for (RSOS-190979.R0)

See Appendix C.

Decision letter (RSOS-190979.R1)

01-Aug-2019

Dear Dr Biazzo,

I am pleased to inform you that your manuscript entitled "General scores for accessibility and

inequality measures in urban areas" is now accepted for publication in Royal Society Open Science.

on behalf of Prof Miles Padgett (Subject Editor)
openscience@royalsociety.org

Follow Royal Society Publishing on Twitter: [@RSocPublishing](https://twitter.com/RSocPublishing)
Follow Royal Society Publishing on Facebook:
<https://www.facebook.com/RoyalSocietyPublishing.FanPage/>
Read Royal Society Publishing's blog: <https://blogs.royalsociety.org/publishing/>

Appendix A

Comments to Author:

Reviewers' Comments to Author:

Reviewer: 1

Comments to the Author(s)

Overall, I think this is a very interesting paper. This paper makes a unique contribution in offering a new method in measuring temporal accessibility and its relevance with social equity. The platform created provides excellent examples of visualizations of temporal accessibility in several world cities and has potential implications for transport policies, particularly in transport equity. Other cities may easily create similar accessibility maps using the codes provided by the authors.

Reviewer: 2

Comments to the Author(s)

One of the interesting things about the “open review” process at the Royal Society Open Science journal is the idea of an ‘unmasked’ peer review: where I can explicate my own academic personality as a background to my reviewer judgment. So to start with that, I am Dr. Michiel van Meeteren, lecturer in human geography at Loughborough University. The reason why I believe I was chosen as a reviewer to this paper is because I was a co-author on the following paper:

“Boussauw, K., Van Meeteren, M., Sansen, J., Meijers, E., Storme, T., Louw, E., Derudder, B. & Witlox, F. (2018). Planning for agglomeration economies in a polycentric region: Envisioning an efficient metropolitan core area in Flanders, *European Journal of Spatial Development*, 69. <http://doi.org/10.30689/EJSD2018:69.1650-9544>”

In this paper we combine accessibility analysis with isochrone maps for a practical application in public transport planning. In doing so we explicitly draw on the long established history of using these techniques in transport geography and related disciplines. I have to admit up front that the method proposed in this paper is more sophisticated as what we used, and I am actually quite enthusiastic about the indicators that this paper proposes. However, my disciplinary background is also a source of criticism in this review, in the sense that “the claims to newness” in this paper does disenfranchise the contributions from geography and geographers to the study of accessibility somewhat. Claims that this paper is “a first contribution in this direction” (abstract), that “studying cities in terms of travel time is a minoritan view” (page 2, for geographers it is the default view). “very few people have studied the temporal features of these systems” (page 2). tie into this feeling. It might be new to complexity science, computer science or physics, but it is not new to geographers at all. This is something the authors acknowledge when they cite some of geography’s foundational works, references 21-29 contain some of the classic papers I would normally recommend. This resonates with a broader annoyance as data science, physics and complexity theorists have been ‘discovering the city’ as a topic of research. We as geographers welcome new perspectives and input to some of our established research topics, but we do like to have some acknowledgement of our historical contribution to a topic, a recurring complaint nicely elaborated in O’ Sullivan and Manson (2015).

We changed the phrase in the abstract:

“This paper gives a first contribution in this direction by providing a new perspective on how to evaluate accessibility in cities based on data of public transportation.”

“This paper gives a contribution in this direction by providing general methods to evaluate accessibility in cities based on data of public transportation.”

We changed the phrase at line 29:

“To this end, we start from a relatively minoritarian point of view, namely that of measuring cities in terms of travelling times rather than geographical distances.”

“To this end, we measure cities in terms of travelling times rather than geographical distances.”

Wh add a sentence, add a citation [...1] at line 70:

“The proposed metrics allow for a capillary study of the level of accessibility of urban areas, a concept formulated several decades ago, and largely used in different context in the literature [...],1...”

[1] Boussauw K, Van Meeteren M, Sansen J, Meijers E, Storme T, Louw E, et al. *Planning for agglomeration economies in a polycentric region: Envisioning an efficient metropolitan core area in Flanders*. 2018

The authors mention that “that the science of cities is a relatively new research area (P.2)”. I would make the case that it needs qualification. I’d say that the author of that claim, Michael Batty has been doing the science of cities for the whole of his long and distinguished career and that he himself was able to establish that career on a network of scholars in the 1960s (Barnes and Wilson 2014 provide historical perspective; and I did my little bit in Van Meeteren and Poorthuis 2018). In fact, accessibility analysis goes back to the collaboration of a geographer and a physicist on population potential in the 1950s (Stewart and Warntz, 1958). In other words, this reviewer would be very happy to see this contribution not as the “invention” of a debate/concept but as a “contribution/refinement” to a longstanding, but perhaps at certain times in history dormant research agenda. Most of this revision is just a question of language and due citation. Even nowadays there is quite a bustling community of geographers working on these topics, for instance just look at journals such as “Environment and planning B: urban analytics and city science” (edited by Mike Batty), The Journal of Transport Geography or Transportation Research. So although I agree with the authors that big data and massive computation offers great potential to revisit these topics (page 4), it is not that there is not a large community of scholars doing exactly that.

We changed the following sentence at line 12:

“The science of cities is a relatively new research area that greatly benefited, in the last decades, of the digital revolution”

“The science of cities is a research area that greatly benefited, in the last decades, of the digital revolution”

Which brings me to another comment: I don’t think this community of geographers has no standard measures of accessibility: we have (see Boussauw et al 2018 for references) and they are quite similar in type to what this paper proposes.

That said, I am very enthusiastic about the new measures “velocity score”, “Sociality score” and “city sociality” and I like the crisp and unambiguous way in which the authors define it. Those measures will be a very welcome and useful tool for comparative research across cities both for academic and for practical applications. Geographers do not like describing indicators with adjectives like “universal” or “law” for all sorts of epistemological and historical reasons (Barnes 2013). Nevertheless I understand there is a problem of translation between disciplinary languages here.

We added the following sentence at line 68:

“The metrics adopted are defined “universal” in the sense that they can be applied in every city and in different contexts allowing comparison between different areas and means of transportation.”

To conclude there is one final weaker point in the paper I would like to point out. The section “Space-time distribution of inequality in accessibility patterns” (page 9-10) assumes a monocentric model of a city. This monocentric model has been increasingly found to be unrealistic of city structures regardless of where you are (Van Meeteren et al. 2016 is a good introduction to these debates see also Clark and Kujipers-Line 1994, Clark 2000). Of course there are car based and public transport based cities and North American cities tend to be the former for historical reasons. So the rankings provided do not surprise a geographer at all. They are exactly what one would expect based on geographical knowledge. However, if the paper is serious in its ambition to provide a cartographic tool that helps policy makers, it should take into account that cities have several employment centres and that the traditional ‘urban core’ has long since lost its meaning as the principal centre of employment public transport is directed toward. Most new jobs come in the city’s edge (Phelps, 2017; Keil, 2017), therefore championing cities that have the best accessibility to the city center is not always the best policy advice, although advocating for higher densities that allow for public transportation surely is. On that note, there is a literature on ‘threshold values’ when public transport becomes viable. This is not so much a “general mechanism” (page 10) as the simple result of transport economics that require a minimal number of customers for an investment to be politically and economically sustainable. Again here we encounter some disciplinary lost in translation. For social scientists these “general mechanism” are the product of social and political decisions which makes us cautious in using terms like “general” and “universal” as society could easily change that reality if enough people want to.

In our analysis, we do not assume a monocentric model of a city. We start only from the point with the highest value (called “center”) of the accessibility quantity under exams and we see how the quantity varies with the time distance from that “center”. What we observe is a fast decay of the average quantities from the “center” with the time distance. It is also important to highlight that our findings are not in contrast with a “polycentric” model of cities. As a matter of fact, our observations could be in agreement with a polycentric city, where the centers are close to each other in public transport travel time.

We changed the following sentences, in the subsection “Space-time distribution of inequality in accessibility patterns” at line 370, in order to reduce possible misunderstandings:

“The maps shown in Fig.2 highlight the fact that the best performing public transport areas are typically clustered in the center of the cities, while many other areas experience poorly performing public transport. To better quantify this effect we display the behaviour of the velocity and of the sociality scores (Fig.8) as a function of the travel-time from the center of each city. Here the center is defined for each city as the hexagon with the highest score (velocity and sociality, respectively).”

“In the maps shown in Fig.2} (and at www.citychrone.org) we observe a central area with the highest values of the accessibility observables and some “islands” with high accessibility values connected to the central zone by some well-served directions, consistent with the idea of polycentric cities [41]. To better quantify this effect we display the behaviour of the velocity and of the sociality scores (Fig.8) as a function of the travel-time, instead of using the spatial distance, from the “center” of each city. Here the “center” is defined, for each city, as the hexagon with the highest score (velocity and sociality, respectively).”

Though we understand that some of our results could be expected at a qualitative level, we think our paper could make a good contribution towards quantification of what can be intuitively guessed; for instance, to quantify the level of inhomogeneity of specific observables on different cities and different areas. These observables could be very useful, for instance, to check the validity of specific assumptions and modelling schemes about cities.

Lastly, the ‘most of the world now lives in cities’ argument is now so much of a cliché (that hides a lot, see Brenner and Schmid 2014) that I would avoid using it.

We change the sentence:

“By the end of this century, most of the world population will be living in cities. The unprecedented level of urban interactions and interconnectedness represents a big challenge to manage the unavoidable growth while aiming at sustainability and inclusiveness.”

“In last decades the acceleration of the urban growth led to an unprecedented level of urban interactions and interconnectedness. This represents a big challenge to manage the unavoidable growth while aiming at sustainability and inclusiveness.”

I hope these points allow the author to improve on what is already a really interesting paper. I look forward to seeing it develop further.

Barnes, T. J. (2013). Big data, a little history. *Dialogues in Human Geography*, 3(3), 297–302.

Barnes, T. J., & Wilson, M. W. (2014). Big Data, social physics, and spatial analysis: The early years. *Big Data & Society*, 1(1), 1–14.

Boussauw, K., Van Meeteren, M., Sansen, J., Meijers, E., Storme, T., Louw, E., Derudder, B. & Witlox, F. (2018). Planning for agglomeration economies in a polycentric region: Envisioning an efficient metropolitan core area in Flanders, *European Journal of Spatial Development*, 69.<http://doi.org/10.30689/EJSD2018:69.1650-9544>”

Brenner, N., & Schmid, C. (2014). The “Urban Age” in Question. *International Journal of Urban and Regional Research*, 38(3), 731–755.

Clark, W. A. V., & Kuijpers-Linde, M. (1994). Commuting in restructuring urban regions. *Urban Studies*, 31(3), 465–483.

Clark, W. A. V. (2000). Monocentric to Polycentric: New Urban Forms and Old Paradigms. In G. Bridge & S. Watson (Eds.), *A Companion to the City* (pp. 141–155). Malden/Oxford: Blackwell Publishing.

Keil, R. (2017). *Suburban planet*. John Wiley & Sons

O’Sullivan, D., & Manson, S. M. (2015). Do Physicists Have Geography Envy? And What Can Geographers Learn from It?. *Annals of the Association of American Geographers*, 105(4), 704–722.

Phelps, N. (2017). *Interplaces: An Economic Geography of the Inter-urban and International Economies*. Oxford University Press

Stewart, J. Q., & Warntz, W. (1958). Macrogeography and Social Science. *Geographical Review*, 48(2), 167.

Van Meeteren, M., Poorthuis, A., Derudder, B., & Witlox, F. (2016). Pacifying Babel's Tower: A scientometric analysis of polycentricity in urban research. *Urban Studies*, 53(6), 1278–1298. <http://doi.org/10.1177/0042098015573455>

Van Meeteren, M., & Poorthuis, A. (2018). Christaller and “big data”: recalibrating central place theory via the geoweb. *Urban Geography*, 39(1), 122–148.

Reviewer: 3

Comments to the Author(s)

If my assumptions are correct, then the methodology is sound. However, the use of imprecise scientific terminology and the lack of rigour when explaining the mathematical equations often forces the reader to guess what the authors meant instead of what they wrote. This becomes particularly problematic when it results in masking some important limitations of the methodology. Although I believe that the following points should prove relatively easy to fix, I would strongly advise against publishing the manuscript until these are addressed, hence the major revision recommendation.

(1) L. 206: The speed of expansion of the isochrone is dr/dt , not r/t . Please indicate the correct interpretation (l. 208-209) directly. In addition, the term “velocity” is generally used instead of “speed” when the direction is considered, while it has precisely been removed here.

We changed the following sentences at lines 204, 210,211, 213:

“we are interested in the ~~velocity~~ average speed of expansion of the front of the isochrone as a function of time.”

“we obtain a ~~“circular” velocity~~ quantity that has the dimension of a speed”

“is the ~~velocity~~ average speed of expansion, at time t , of a circular isochrone with the same area of the real one”

“This quantity can be thought of, approximately, as the average velocity of a journey of duration t choosing a random direction from a starting point.

(2) L. 209-210: v is certainly not proportional to the explorable area from the hexagon λ . If an isochrone moves at a constant speed of 1 m.s⁻¹ and is a perfect circle, then $v(1)=1$ and $A(1)=\pi$, while $v(2)=1$ and $A(2)=4.\pi$, and more generally, $v(n)=1$ and $A(n)=n^2.\pi$ for any n . The authors probably meant that for different λ s in one or more cities and for a common τ , then v is proportional to the explorable area. The text needs to be changed accordingly.

We changed the following sentence at line 215:

“On the other hand, this quantity is proportional to the square root of the amount of area it is possible to explore from the hexagon λ given a time interval t ”

(3) L. 235-236: $A(\tau)$ for a typical daily τ is already a better measure than v from eq. 3 of “the extension of the area that it is possible to explore in a typical working day”. Meanwhile, v is the average speed at which an individual should expect to travel on their typical daily trips. Why is the first interpretation chosen rather than the second?

We chose to consider the square root of the area, divided by the time, because this quantity, in an approximate way, can be seen as the average velocity from the point considered taking a random direction of displacement. We have the feeling that, when speaking about transportations, is easier for all to quantify and imagine a speed rather than an area as a function of time, so it is easier to make a comparison with our everyday life and communicate the scientific results to a large audience with very different backgrounds.

We changed the sentence at line 218:

“We chose to consider the square root of the area instead of the area itself to have a more direct interpretation of it in terms of transportation velocity, **that is easier to communicate and to understand for a general audience.**”

(4) Eq. 3: I am guessing from the supplementary information and from the legend of fig. 2 that “N is a normalisation constant” means that its role is to ensure that $\int f(t) dt = 1$ and this would explain why the division by $\int f(t) dt$ has been omitted in eq. 3. This omission should be properly indicated, otherwise the dimension of v is not $[m].[s]^{-1}$, but only $[m]$ and the integral becomes a sum rather than an average. More details about $f(t)$ and at the very least a reference to the supplementary material should appear in the main text. Also, presenting $f(t)$ as the direct result of a survey is misleading, since it is instead the result of some modelling work using an external methodology (which comes with its limits, see (6)).

We changed the equation Eq. 3 and Eq. 4 in order to ensure that the dimension of the average is $[m][s]^{-1}$. We added explicitly the normalization.

We changed the following sentences at line 224:

“The distribution adopted is taken from **an analytic curve obtained from fits of surveys of the daily budget times spent on a bus by UK citizens [1]. In the Supplementary Material File S1, it is shown how the accessibility measures proposed are robust against reasonable choices of travel time distributions and the results presented do not sensibly change by choosing different distributions.** Fig.3 (panel A) shows the velocity scores of six different cities. For interactive explorations of the maps and other cities we refer the reader to the platform `\url{citychrone.org}`. ~~In the Supplementary Material File S1 it is shown how the accessibility measures proposed are robust against reasonable choices of travel time distributions and the results presented do not sensibly change by choosing different distributions.~~”

(5) The description of $f(\tau)$ in the supplementary material is still hard to understand. What is the value and role of N? What is the quality of the fitting (e.g. R^2)? Why does T_{bus} mention only a bus when the survey is now about “transport habits” compared to “Oyster card journeys on bus, Tube, DLR and London Overground”?

We used empirical travel-time distributions that we were able to find in literature. We could find only two analytic forms derived from empirical data in the literature. Then we add a third one, the “flat” distribution, just to check if our results are sensitive to the choice of the travel time distributions. What the Fig.1 of the Supporting Material file S1 shows is indeed that our results are robust against a different choice of travel-time distributions. The “N” is a normalization constant. It ensures that the distribution sums to 1.

We added the following sentence in the supplementary:

“where N is a normalization constant that ensures that $\int_0^{\infty} f_{\text{DBT}}(t) dt = 1$.”

(6) The study on which the $f(\tau)$ distribution is based only uses (significantly outdated) data from the UK. As a matter of fact, the authors of the original study lengthily discuss the limitations of their data. The only improvement made is a comparison to some more recent data that already shows discrepancies despite being from the same geographical region (which is not even featured in the case studies). This is an important flaw that should be mentioned in the discussion in the main text of the manuscript.

As stated earlier, our results are quite stable against a reasonable choice of the travel-time distributions, meaning that different choices of those distributions, did not affect our results.

(7) L. 236-237: It would be useful to indicate that people are counted multiple times. For example, a typical trip from inside inner Paris is likely to stay inside inner Paris which only has a population of 2M, while fig. 2 suggests a typical traveller would reach up to 3M people in a day. On a side note, I would replace “meet” by “reach” on l. 237.

In our approach, people are not counted multiple times. We use the definition of an urban area as *economic functional units* given by OECD/EU, as written in the main text, in the Methods section. In the case of Paris, the urban area contains more than 12 millions of people. Due to the high level of development of the public transports, from the center of Paris, one can reach more than 3 millions of people, some of them, of course, living outside the municipality of Paris, though still within its urban area.

(8) L. 300: “without lack of generality” is a strong claim considering that this subset of cities contains only two clusters of very similar cities.

We changed the sentence in the following way at line 308:

“We focus, ~~without lack of generality~~ for sake of visualization, to a subset of cities, namely the same cities we focused on in the section devoted to Accessibility Metrics: Paris, New York, Madrid, Montreal, Sydney, Boston. In the Supplementary Material File S1 we show how the results presented are valid also for the other cities analyzed.”

(9) L. 315-320: Please discuss the varying cultural expectations regarding car usage. Inequality in accessibility through public transport does not necessarily mean inequality in practical accessibility if the use of cars has been assumed a priori by urban planners.

Our analysis is only about the performance of public transports in the city. The accessibility quantities defined, although can be used also for car displacements, have been used to measure the public transports performance. All the statements made about the accessibility in the cities analyzed are referred to public transports.

In the section “Inequalities in urban accessibility patterns” we study the inequalities in the access to the public transports among populations, not the absolute values. The point is that we observe common patterns among cities regardless if the urban planners design the city to be more cars or public transports oriented.

We change the sentence in the presentation:

“In particular, we show that while the distributions of the accessibility metrics of public transports seem to have higher values for high-density areas, yet only a small fraction of the population have access to high-performing public transportation means.”

And the following sentence at line 324:

“A very inclusive city has this distribution peaked around high values of the Velocity score. From this perspective, Paris, New York and Madrid appear to be more inclusive than Montreal, Sydney and Boston”

“A city, with well-distributed public transport accessibility among the population, has this distribution peaked around high values of the Velocity score. From this perspective, Paris, New York and Madrid appear to have more equally distributed velocity score than Montreal, Sydney and Boston.”

(10) L. 341-354: This result is an artefact due to the combination of some hexagons being directly on the fastest lines together with a daily trip distribution that only represents the “average habit” over the entire city. People who are more “travel averse” will tend to pay a higher rent to be closer to the fastest lines, specifically to limit their “travel energy budget”, without necessarily belonging to the 1% richest. An interesting analysis could have consisted in linking the hexagons with the social characteristics of their inhabitants and checking if the most socially privileged are also the most privileged in accessibility. Failing to do that, at least discuss the limits of this “1% of the hexagons” approach.

Our analysis is trying to measure the performance of public transports to cover the space and connect people regardless of the “habit” of the people. As previously explained, our results are quite stable against a reasonable choice of the “average habit” choose. We do not consider the 1% richest and the rest 99%. We only consider people or area with accessibility 1% higher in comparison to the rest 99%.

The analysis of the connections with the economic, cultural or habits diversity will be the subject of future studies.

(11) L. 78-81 & I. 404-406: These statements are very subjective. It is a normal part of science to study a notion in all its aspects. As a matter of fact, a broader perspective on the subject would have been beneficial to the quality of the “Inequalities in urban accessibility patterns” section of the manuscript, as illustrated in the two previous points. Please rephrase (l. 78-81 in particular) in a more neutral and more moderate way.

We changed the sentence (l.78-81):

“This proliferation of definitions ~~is definitely not helping~~ can make it difficult in reaching a unifying view about cities and their dynamical aspects, ~~contributing instead to a dispersion of scientific efforts in diverging directions.~~ Moreover, the lack of a comprehensive and easy to understand definition of accessibility, could prevent policymakers from using it in an operational way and scholars from comparing different approaches and methodologies [1].”

We removed the sentence (l. 404-406):

“Many different metrics of accessibility have been proposed, whose scope of application is often restricted to the specific context for which they have been introduced.”

[1] Geurs, Karst T., and Bert Van Wee. "Accessibility evaluation of land-use and transport strategies: review and research directions." *Journal of Transport Geography* 12.2 (2004): 127-140.

Appendix B

Dear Editor,

Thanks a lot for granting us the possibility to re-submit our paper as well as for the possibility to reply to the remarks and comments made by Reviewers 2 and 3. We take the opportunity to thank the three experts who reviewed our paper. Here is an additional list of potentially new reviewers:

- Michael Batty, UCL, London, UK
- Jose' Ramasco, IFISC Palma de Mallorca, Spain
- Marta Gonzalez, UC Berkeley, US
- Laura Alessandretti, Center for Social Data Science, Copenhagen, Denmark
- Marc Barthélemy, CEA, Paris, France
- Laetitia Gauvin, ISI Foundation, Italy
- Michele Tizzoni, ISI Foundation, Italy
- Alessandro Vespignani, Northeastern University, US

Below are our detailed replies to Reviewers 2 and 3.
We hope the review process can now proceed swiftly.
Our warmest regards,

Indaco Biazzo, Bernardo Monechi, Vittorio Loreto

Response to Reviewer 2:

We thank the reviewer for his/her comments and support to our work. His/her contribution was extremely valuable in helping us to better shape our text to make it accessible also to a general audience outside our field of research. The reviewer's comments are addressed in the following text.

1: One overarching comment is that the paper would benefit from a thorough language edit. Although everything is understandable, there are repeated minor grammatical and spelling errors and many dangling modifiers (where the reference object in the previous sentence is unclear), and repeated language that could be avoided by use of synonyms. A language specialist might choose different adjectives / phrases in order to get a more nuanced perspective across, and sentences could be split to improve readability. For instance, the first sentence of the introduction is more than four lines, which is not very common in the English language (contrary to many Latin languages). "Lost in translation" issues could also be in play in some of the other minor things I raise below.

We revised the manuscript accordingly, trying to remove the grammatical and spelling errors and simplifying and shortening sentences.

2: What do you mean with "the fruition" of metrics? (page 12, line 32). I agree that your metrics are intuitive, but do you mean "making their practical application easier" or something like that?

Indeed. We meant that our metrics and results should be easily reproducible by other researchers. We modified the sentence in the following way: "Also, our metrics are well suited for being shared and easily visualized on maps, making them easy to be applied by other researchers to reproduce and extend our results."

Both in the abstract and on pages 18-19 there is the claim that there is a "general mechanism" or "common mechanism" at work that can possibly explain the regular patterns of inequality found. I would advise some caution here. The notion of "mechanism" suggests ideas about causation of spatial inequality which is somewhat outside the scope of the current paper. I wholeheartedly agree that there is a recurring pattern across cities, and that this seemingly "universal pattern" merits further investigation. However, what causes these patterns is a question that is yet to be answered. Even if one was able to describe the mechanism in a morphological sense (for instance through a process of preferential attachment) it still could have different causes. Some may be described in the realm of transport economics (minimum densities to profitably exploit public transport or alternatively through network externalities), labour market geographies (places of work and places of living), theories of rent and so on. I would suggest being slightly more modest and claim that there is a recurring morphological pattern of inequality in accessibility across cities that is very interesting to disentangle in future urban-scientific research. But like my similar remark in the previous review round: I am aware that this is viewed differently across disciplines, and that where I come from we tend to be very cautious when it concerns causality.

We agree with the Reviewer that we are not investigating the causes of the observed patterns. Accordingly, in the abstract we removed the reference to a "common mechanism", simply pointing out the similarities across cities:

"We highlight great inequalities in the access to good public transport services across the population, which are found to be strikingly similar across different towns."

As the reviewer pointed out, the adoption of the term "mechanism", when referring to the observed patterns, may be misleading. Still, what we meant was that, even though the causes of those patterns have yet to be elucidated, we conjecture that similar causes could lead to similar outcomes. In order to be well balanced, we changed the text along the following lines:

"These results exhibit strongly similar patterns among all the observed cities, suggesting the existence of similar causes underlying the observed phenomenology, that could range from morphological to Socio-Economic ones."

We also turned "a general mechanism" into "shared causes independent from the particular location".

In the discussion section, we speculate a bit about such causes, also following the comments of Reviewer 3 about the adoption of private cars and the need to connect important points with limited resources:

“The observed similarities of the mobility patterns across different cities suggest the existence of common causes, independently of the specific location. The observed inequality patterns are the results of the planning and organization of public transport systems. Though disentangling such causes remains a challenge, one can speculate that these patterns might emerge as a consequence of the limited resource urban planners have to deal with when designing public services. Under these circumstances, planners may be forced to secure first efficient services to highly important locations, leaving behind less crowded areas or less economically crucial areas. In doing this, planners might foresee the use of other transportation means (e.g., private cars), thus leaving certain areas with poor public transport connections.”

Response to Reviewer 3:

We thank the review for the constructive comments on our work. In the following, we hope to address the issues the Reviewer pointed out in a satisfactory manner. Despite some misunderstandings and limitations, we still think our work is relevant for the field of spatial accessibility studies and we hope the Reviewer will be convinced as well at the end of this round of review. The first issue raised by Reviewer 3 states:

“Claiming that the Supplementary Material ascertains the robustness of the methodology is misleading, if not dishonest.”

The accusation of being misleading, or worst dishonest, though certainly a bit over the line, can be due to a misunderstanding that we hope to clarify here. “Robustness” in our case means that the results are qualitatively independent from the travel time distribution used to compute our accessibility metrics. Either if we use the empirical travel budget distribution or another synthetic one, we do not observe large variation in the results. An area with a relatively large velocity score will have a large velocity score changing travel time distribution.

The main criticism about the methodology is described by Reviewer 3 in point 2 of his/her second review:

Point 2.

Comparing two travel time distributions obtained for one city, then applying these distributions to completely different cities is the exact opposite of a “robust” approach. Where is it justified in the manuscript that the empirical daily budget for London residents is relevant to Boston or Sydney? Why should we expect people living in Florence or Toulouse to devote as much time to transport as people living in a city 15 times bigger? In addition, what is the quality of these fittings compared to real data, not compared to other modelled data?

We understand the concerns raised by the Reviewer, about the choice of the travel time distribution. We thank him/her for raising this important remark that allows us to better clarify our approach and methodology. We agree with Reviewer 3 that the travel time distribution could vary between cities, and also among groups of different cities, and at a microscopic level from an individual to another. Our aim is that of measuring the performance of public transports to connect places and people **from the point of view of a single or a group of individuals**. This is why we use the same travel time distribution for every point in every city. In general, we agree with the Reviewer that our accessibility metrics quantitatively depends on the specific travel time distribution used. Still, how our accessibility metrics are defined as guarantees that the results presented do not change qualitatively upon changing the travel time distribution. In particular, the ranking of cities as well as the inequalities pattern observed do not change, as reported in the main text as well as in the Supplementary Material. From this perspective we consider our results robust and independent from the specific travel time distribution used and in general from the habit of people. More in general, the adoption of accessibility metrics computed by keeping fixed the travel time distribution, could be suitable to single out areas better served by public transportation in any city from an individual point of view. In order to clarify this point we added in the Results section the following paragraph:

“In equation (\ref{eq:vel_score_avg}) the average over τ is performed by weighting with a travel-time distribution $f(\tau)$. The travel time distribution represents the probability for an individual or a group of individuals to perform a journey of duration of τ . The travel time distribution could vary between the considered cities, time frames~\cite{geurs2004accessibility}, and also between areas and groups of individuals of the same city~\cite{Accessibility_Houston}. In the Supplementary Material File S1 we show how the \emph{Velocity Score} (and the other accessibility metrics defined in the following) computed with different choices for $f(\tau)$ are highly correlated with one another. Thus, the choice of $f(\tau)$ does not alter qualitatively the results obtained. On the other hand, using the same $f(\tau)$ for each city is equivalent to focusing on the perspective of a single individual, or a cohesive group of individuals, who would compare different cities and different transportation systems from their perspective. For all these reasons we focused on one specific travel time distribution, namely that obtained from fits of surveys of the daily budget times spent on a bus by UK citizens~\cite{kolbl2003energy}. We remark that, though out of the scope of the present paper, the investigation of the impact of different city-specific travel time distributions would deserve a research effort.”

Point 1.

I agree. Hence, I don't understand why the first clear and neatly written interpretation of $v(\lambda)$ (given I. 238), the one people are likely to remember, is “the Velocity Score provides with a measure of the extension of the area” rather than the speed interpretation.

We apologize for having missed this remark. We agree with the Reviewer. We accordingly modified the text as follows:

“Considering a typical working day, the Velocity Score approximately provides with a measure of the average speed at which an individual can move away from a hexagon

λ , in a randomly chosen direction. The Sociality Score provides instead a measure of the number of people it is possible to reach within the same trip.”

Point 3.

The methodology assumes that the daily budget is entirely spent in one round trip. If, for example, an individual undertakes a first trip from inside Paris to their workplace 4 km away and a second trip to a shop 2 km away, their movements remain in fact bounded within a disk of radius 4 km, representing 64% of Paris, or roughly 1.3M inhabitants. However, the method would count them as traveling up to 6km away, reaching roughly a total of 3M inhabitants. In this case, the 320k inhabitants of the area of the second trip, already counted once in the first trip, add an extra 1.7M to the total count of people reachable. This needs to be properly indicated.

We think there is a misunderstanding here. We are interested in the travel time distribution, not in the budget time distribution. We do not assume that “the daily budget is entirely spent in one round trip”, but we consider that the daily budget distribution is spent in two trips as explained in the Supplementary Material. This is of course a rough approximation. This point was explained in the first page of the Supplementary Material. In order to better highlight this point, we rewrote the section “Robustness of the accessibility metrics definitions with respect to travel time distributions” in the Supplementary Material, where we tested three different travel time distributions.

There are already one unrealistic isotropy assumption (contradicted, for example, by the lack of willingness to cross the “périphérique” - or purpose to do so - for inhabitants of the municipality) and one incorrect assumption that the daily budget is uniformly distributed inside the city (people living inside the “périphérique” do so at a very high cost to avoid spending time in transports). As a result, the sociality score obtained for inner Paris residents is completely unrealistic. In fact, the maps in panel B of figure 2 appear to be a less precise and less accurate version of this map, which provides simple objective information without making any claim of representing anything “typical”:

<http://www.urbanmorphologyinstitute.org/france/>

We would like to thank the Reviewer for signalling this interesting map, which is indeed very related to our work. We honestly do not have a detailed knowledge of the behaviour of the inhabitants of Paris. The inclusion of this element as well as the corresponding one for other cities will certainly be an added value for future works. Concerning the isotropic assumptions, this is typically made for sake of simplicity in many accessibility studies, at least concerning the direction of the movements of the citizens. Such assumption is indeed made also in “<http://www.urbanmorphologyinstitute.org/france/>”, where our same notion of isochrones is considered. The main difference we can see is that such map does not consider any area of the city as equal, but uses as relevant only those in which workplaces

are available. This issue has been pointed out by the Reviewer as well in one of the comments that follows.

The purpose of the manuscript is to propose a “unifying definition of accessibility” that would be directly useful to policy makers. Ignoring how urban planners design their cities does not sound like a good starting point to achieve that. I fail to see why it is not possible to add one or two lines pointing out the role of cars as an explanation to the difference in the observed results between cities.

We understand the concerns of the Reviewer about the use of private cars and the role of private transportations in urban planning. The scope of our work is mainly related to public transportation systems, yet some of the observed patterns might be the result of plans in which both public and private transportation systems are considered. At present, we cannot prove or disprove the conjecture made by the Reviewer that the differences observed in the accessibility patterns of different cities are due to the presence of private cars. Anyway, to open to further studies along these lines, we added the following text in the Discussion section:

“The observed similarities of the mobility patterns across different cities suggest the existence of common causes, independently of the specific location. The observed inequality patterns are the results of the planning and organization of public transport systems. Though disentangling such causes remains a challenge, one can speculate that these patterns might emerge as a consequence of the limited resource urban planners have to deal with when designing public services. Under these circumstances, planners may be forced to secure first efficient services to highly important locations, leaving behind less crowded areas or less economically crucial areas. In doing this, planners might foresee the use of other transportation means (e.g. private cars), thus leaving certain areas with poor public transport connections.”

The “1% highest accessibility hexagons” is not necessarily meaningful. An individual living in the south of Horley (42 km from London) will fare very well since they will be in the same hexagon as the Gatwick express (only £18/trip) and the fast suburban Southern and Thameslink trains. Meanwhile an individual living on Westbourne Grove in zone 1 may (depending on the hexagon placement) have to rely on buses or walk to another hexagon to take the tube.

Despite limited to a max of 15 minutes on foot travelled at 5 km/h, we consider the possibility to walk to a nearby hexagon to access public transport, either at the beginning of the trip or whenever there is the need to change means of transport. The time spent walking is included in the time of the trips. The limit of 15 minutes is again arbitrary, but it can be easily varied in our analysis.

We clarify this in the text:

“Each one of these shortest-time-paths will consider all the possible means of transport between two hexagons, including the possibility to move on foot to nearby hexagons to access the public transport services places within a given area.”

By construction, building a transport network requires to fill a 2-dimensional space with 1-dimensional lines. As cities become larger, this is less and less possible, so planners only try to connect people to places that matter to them. They try to cater to people’s habits, no matter the number of random strangers they can wave hello to from the window of their trains as they pass by towards their intended destination. This is similar to the difference between degree centrality and Katz centrality in network theory. What matters most: the number of nodes one is connected to or the importance of the nodes one is connected to?

We agree with the Reviewer on the fact that the patterns we are observing could result from the constraints urban planners have to face. Being the explanation of these mechanisms beyond the scope of the present work, we just mention this issue in the Discussion section (see comment above)

Some may think that taking these considerations into account participates in the “proliferation of definitions that disperse the scientific effort”. However, the “top 1%” analysis presented in the manuscript is misleading and does not add any valuable “operational” information. It does not reveal anything about inequality in “access to opportunities”, only that central inhabitants could easily go check if the birds are different in the suburban residential areas if they wanted to and that airports and amusement parks are the best spots for high accessibility unauthorized camping.

Here the Reviewer highlights one of the drawbacks of our approach: not considering the distribution of opportunities within the cities. The reason we did not take them into account is purely practical. As for the travel-time distribution, finding these data for many different towns requires a strong effort that we would like to devote to future researches. Nevertheless, we can already argue that the possibility to reach even remote and uncommon locations is correlated with the possibility to reach more important ones easily. Still, it will be essential to include points of interest and opportunities in future works, which, by the way, is quite straightforward in our approach. For example, we could consider the number of recreational areas, of shops of a specified kind or of workplaces instead of individuals reached in our Sociality Score metrics and obtain accessibility maps conveying different information. We modified the text along the following lines:

“The population can be considered as a proxy for the number of opportunities within an area. Given the availability of data, the Sociality Score can be easily modified to consider only opportunities of a specified kind. It is sufficient to substitute in equation $(\text{ref}\{eq:sociality_score\})$ the amount of population within the isochrone with the amount of that kind of opportunities.”

And in the discussion:

“Our framework is easily extendable to clarify these issues since the inclusion of other data sources concerning private transport and the inclusion of points of interest in the accessibility metrics (e.g. workplaces, shops, schools, etc.) is quite straightforward.”

Appendix C

Response to Reviewers

We want to thank the editor for having accepted to assess our paper, based just on one reviewer's comments. We addressed the concerns about the English language polishing by having it revised by an English native speakers, working at Sony CSL Paris. Michael Ainslow (michael.ainslow@sony.com) is familiar with scientific english and has revised our manuscript, correcting and rephrasing some of its parts.

The editor is free to contact him and he will provide proof of their involvement in the revision of the paper, as well as a report of his corrections.

We would also like to thank the reviewer for his valuable comments. We agree with his concerns about the statements made in the paper, which are due to our different backgrounds, and we are more than willing to solve. The reviewer gave us a precise set of points to be addressed. Hence, we will first discuss his comments and then provide the changes we made in the text for each provided point.

All the sentences added and corrected are highlighted in red in the manuscript. We indicate them in the following with the part of the text in which they can be found.

Reviewer 4

I believe the paper does make a contribution to urban accessibility studies by proposing indices that can be used to compare pedestrian and public transit accessibility levels from city to city. I The three indices proposed in the paper – velocity, sociality, cohesion – are complementary. I think the paper deserves publication, but there are still a few issues the paper should address before publication, however.

First, as has already been indicated in previews reviews, a thorough language edit is still needed by a native English speaker.

We agree with the reviewer and we have addressed this issue in the response to the editor at the beginning of this document.

Second, the most important issue I find with the paper is the assumption that trips are modeled from all hexagons to all hexagons. I understand the simple appeal of this assumption – which is also common in computer science and graph theory algorithms, where indices are computed from all nodes to all nodes. It offers a simple way to make cities comparable to each other, but it doesn't really describe real commuting flows nor aims of transit systems.

As the reviewer mentioned, this choice is a strong assumption. What we would like to communicate is that our aims are not to describe the real commuting flow or the way public transport are design. Our principal aim is to measure the performance of public transport at connecting places or people in cities. As we better describe below, we add and change several sentences in the text to better highlight this aspect.

Urban populations do not commute with equal likelihood to all parts of the city. Trips are much more common from homes to jobs, homes to commercial destinations as well as homes to recreational destinations and vica versa These destinations types are never uniformly distributed in space – they

tend to concentrate in the CBD and other subcenters. Accordingly, public transportation systems and street networks evolve over time to prioritize access to such destinations instead of maximizing uniform access to all parts of the city. If we measured how easily people from all parts of the city can gain access to jobs as well as commercial destinations, we would likely find different results. The 1% VS 99% access differences highlighted in the paper would likely change. And the meaning of velocity, sociality and cohesion would likely change.

Again, we agree with the reviewer. If we use instead of population, like in the case of sociality score, the distribution of jobs we obtain a different accessibility measure, that may be called for instance jobs score. Up to now we just analyze the performance of the public transport to connect areas and people.

People meet others not at their homes, but in public or commercial and recreational spaces of the city. We should thus not judge the quality of public transit systems based on how well they allow residents to get to all other residents' homes.

We remove all the sentences in the text that could lead to the interpretation that our aims were to assess the quality of public transport in general. We always try to make clear in the text that we want to measure only the performance of public transport to connect people or places. Here there is a list of sentences that were changed in order to reduce the misunderstanding:

Abstract:

- In the last decades, the acceleration of urban growth has led to an unprecedented level of urban interactions and interdependence. This situation calls for a significant effort among the scientific community to come up with engaging and meaningful visualizations and accessible scenario simulation engines. The present paper gives a contribution in this direction by providing general methods to evaluate accessibility in cities based on public transportation data. Through the notion of isochrones, **the accessibility quantities proposed measure the performance of transport systems at connecting places and people in urban systems.** Then we introduce scores rank cities according to their overall accessibility. We highlight significant inequalities in the distribution **of these measures** across the population, which are found to be strikingly similar across various urban environments. Our results are released through the interactive platform: www.citychrone.org, aimed at providing the community at large with a useful tool for awareness and decision-making.

Introduction:

- [line 82] It is worth mentioning that the local nature of our metrics allows us to evaluate and visualize the geographical fluctuations of the velocity and sociality scores, and thus to quantify the inequalities **distribution of these measures among areas and population** within each city. In particular, we show that while the distributions of the accessibility metrics seem to have higher values for high-density areas, only a small fraction of the population **lives in areas with accessibility scores much larger than the rest of the city.**

Inequalities in urban accessibility patterns:

- [line 315] In this section, we focus on a particular aspect of accessibility, **the spatial-temporal distribution inside the city**.
- [line 368] Similar considerations hold for the sociality scores, which implies that 1% of the population potentially has access at twice the number of **people** as the rest of the population.
- [line 377] The quantitative assessment of the strong **uneven distribution** observed in the accessibility patterns reported above can be further clarified by looking at the spatial distribution of the accessibility metrics.
- [line 353] The spread of the distributions is still very high and very few people (or areas) have access to high **accessibility values** compared to the rest of the populations (or areas).
- [line 406] Moving in space and time away from these areas will lead to experiencing generally **much less performing** public transport services.

Discussion:

- [line 438] A very interesting opportunity open by the new scores concerns the possibility to quantify the level of **uneven distribution of these quantities** within a city, i.e., the fluctuations of the accessibility scores among **areas and population**.

We agree with the reviewer that people do not meet others only at home. The sociality score gives also a measure of how many people can reach a place. If we assume that, on average, the travel time of a trip is very similar to the time of the reverse trip, the sociality score can be interpreted as the amount of population can reach a specific location.

The following plot [added in the supplementary material] shows the high correlation between the sociality score of Rome computed with travel time of incoming and outgoing trips.

In order to better clarify the nature of sociality score we add the following sentence in the text:

[line 249] The sociality score can also be interpreted as a measure of the amount of the population that can easily reach the point considered if we assume that, on average, the travel time of trips in cities is similar reversing origin and destination. In order to validate this assumption we compute the sociality score with travel time of the incoming trip and outgoing trips for each point in Rome. In fig.5 in the Supplementary Material File S1 there is the scatter plot of these two quantities showing the high correlation between these two measures.

Finally, we agree with the reviewer that a more detailed understanding of the quality of public transport would require taking into account the distribution of opportunities and points of interest within the city and the travel choices of individuals. As we briefly discussed in the Discussion section, the inclusion of this kind of data would be straightforward in our method and would certainly lead to interesting results. We tried to stress this more in the text. We also tried to clarify that our results are far from being conclusive and more investigation is needed in this direction.

In the following, we follow the suggestion of reviewer 4 and we changed the text accordingly to the issue raised.

To address this issue, I suggest that:

a) *the discussion section should either eliminate or substantially change the discussion about the inequality of access in cities.*

We change the discussion about inequality access in cities. We highlight principally the uneven distribution of our accessibility quantities among population and areas. We change/add the following sentences:

[line 438] A very interesting opportunity open by the new scores concerns the possibility to quantify the level of uneven distribution of these quantities within a city, i.e., the fluctuations of the accessibility scores among areas and population. We remark here that our first aim was to measure the performance of public transport to connect places and people. Despite that more realist origin destination matrix, for instance considering the distribution of opportunities within the city, can be considered and easily integrated into our framework. But, up to now, there is still a lack of open data-sets covering enough cities for this kind of analysis. Taking into account the aim of the accessibility measure proposed, our analyses reveal a general pattern observed in all the considered cities. Namely that the 1% of the area of a city features accessibility scores with average values at least double those of the remaining 99% of areas.

The same patterns are observed by looking at the number of people enjoying specific values of the accessibility scores: also in this case, the top 1% of the population can move at least twice as fast as the remaining 99% of the population. This very uneven distribution of performances of the public transport within an urban environment is explained in terms of the rapid decay of the accessibility scores as a function of the temporal distance from the city center. The observed similarities of the mobility patterns across different cities suggest the existence of common causes, independently of the specific location. The observed inequality patterns are the results of the planning and organization of public transport systems. Considering our initial remarks, we can speculate that these patterns might be explained by the limited resource urban planners have to deal with when designing public services. In this sense, including important locations and fluxes might allow us to understand if these resources are efficiently allocated to satisfy the mobility needs of the citizens.

b) *Some material should be added towards the beginning of the paper about the all hexagons to all hexagons O-D choice.*

We add at the beginning of the accessibility measures section the following sentences:

[line 189] The accessibility quantities proposed aim to measure the performance of public transports at connecting places (velocity score) and people (sociality score). Roughly speaking, the velocity score measures how fast it is possible to reach any point from any other point in the city. The sociality

score measures the amount of population that it is possible to reach from any point in the city. Usually, the flow of people in urban systems is described by an origin-destination matrix (ODM). The velocity score can be thought as an accessibility measure that assumes a uniform ODM. Conversely, in the case of the sociality score, we assume an ODM proportional to the population.

c) *The discussion section should also mention the need in future work to test a similar approach with different assumptions about origins and destinations.*

We add the mention about using different assumptions about origins and destinations flow in the discussion section. The answer to the point “a” shows these sentences.

d) *The paper should reduce its claims that an end-all approach is delivered to compare cities on accessibility levels. Rather, the paper makes an interesting step towards comparable urban accessibility indices but there is a quite a bit more to do before a universal or standardized approach can be agreed upon.*

As explained before we try to better clarify the aim of the accessibility measure we define. In this regard, most of the changed/added sentences listed reduce the claims about the generality of our accessibility measure. Then we add the following sentences in the discussion section:

[line 424] The main aim of this paper is to give a contribution towards a unifying, simple and general framework for accessibility studies. We proposed some general metrics that allow for a quantitative comparison of different cities and different areas of the same city. Despite the limitations of some of our assumptions, our framework and measures are easily reproducible and applicable to the study of accessibility via public transport in every urban environment in which transit feed open data is available.

[line 464] As a final remark, the inclusion of other data sources, such as points of interest in the accessibility metrics (e.g. workplaces, shops, schools, etc.) or considering fluxes of people is quite straightforward in our framework and could lead to interesting results either in the global ranking of accessibility between cities and in the comparisons between city areas, by giving more importance to the purpose and popularity of certain trips.